# CARL: Camera-Agnostic Representation Learning for Spectral Image Analysis

**Alexander Baumann**[1,2,3,*]  **Leonardo Ayala**[2]  **Silvia Seidlitz**[2,4,5,6]
**Jan Sellner**[2,5,6]  **Alexander Studier-Fischer**[7,8,9,10]  **Berkin Özdemir**[8,9,10]
**Lena Maier-Hein**[2,3,4,5,6,†]  **Slobodan Ilic**[1,†]

[1] Siemens AG, Munich
[2] Division of Intelligent Medical Systems, German Cancer Research Center (DKFZ) Heidelberg
[3] Medical Faculty and [4] Faculty of Mathematics and Computer Science, Heidelberg University
[5] National Center for Tumor Diseases (NCT), NCT Heidelberg
[6] HIDSS4Health, Heidelberg
[7] Department of Urology and Urosurgery, University Medical Center Mannheim
[8] Department of General, Visceral, and Transplantation Surgery, Heidelberg University Hospital
[9] Division of Intelligent Systems and Robotics in Urology, DKFZ Heidelberg
[10] DKFZ Hector Cancer Institute, University Medical Center Mannheim

## Abstract

Spectral imaging offers promising applications across diverse domains, including medicine and urban scene understanding, and is already established as a critical modality in remote sensing. However, variability in channel dimensionality and captured wavelengths among spectral cameras impede the development of AI-driven methodologies, leading to camera-specific models with limited generalizability and inadequate cross-camera applicability. To address this bottleneck, we introduce **CARL**, a model for **C**amera-**A**gnostic **R**epresentation **L**earning across RGB, multispectral, and hyperspectral imaging modalities. To enable the conversion of a spectral image with any channel dimensionality to a camera-agnostic representation, we introduce a novel spectral encoder, featuring a self-attention-cross-attention mechanism, to distill salient spectral information into learned spectral representations. Spatio-spectral pre-training is achieved with a novel feature-based self-supervision strategy tailored to CARL. Large-scale experiments across the domains of medical imaging, autonomous driving, and satellite imaging demonstrate our model's unique robustness to spectral heterogeneity, outperforming on datasets with simulated and real-world cross-camera spectral variations. The scalability and versatility of the proposed approach position our model as a backbone for future spectral foundation models. Code and model weights are publicly available at https://github.com/IMSY-DKFZ/CARL.

## 1 Introduction

Spectral imaging, including RGB, multispectral, and hyperspectral imaging, capture channel-wise reflectance information for camera-specific wavelengths. The enriched spectral information, contained in a few to hundreds of channels, enables applications in a variety of fields, including segmentation and classification tasks in medicine (Seidlitz et al., 2022; Ayala et al., 2023), urban scene perception (Theisen et al., 2024; Shen et al.), and remote sensing (Lu et al., 2020; Thenkabail et al., 2018). To develop robust solutions for these tasks, data-driven models have emerged as the prevailing standard, maximizing performance through the utilization of all available images, regardless of camera characteristics. However, the evolution of spectral imaging technology has resulted in significant variability in camera devices (Qian, 2021), leading to the formation of camera-specific data silos. These silos share valuable domain-specific geometric information but differ in spectral characteristics such as channel dimensionality and covered wavelengths. Conventional imaging models such as Convolutional Neural Networks (CNNs) (He et al., 2016) cannot accommodate these

---

*Corresponding Author: `baumann.alexander@siemens.com`,  [†]: Equal Contribution

Table 1: **Comparison of state-of-the-art spectral image encoding approaches.** The proposed model is the only one that incorporates all four desirable characteristics simultaneously.

| Model | Wavelength-awareness | Channel-invariance | Spatio-spectral encoding | Spatio-spectral SSL pre-training |
|---|---|---|---|---|
| SpectralGPT$^+$ | ✗ | ✗ | ✓ | ✓ |
| Spectral Earth | ✗ | ✓ | ✓ | ✗ |
| DOFA | ✓ | ✓ | ✗ | ✗ |
| Copernicus-FM | ✓ | ✓ | ✗ | ✗ |
| SMARTIES | ✓ | ✓ | ✗ | ✗ |
| **CARL (Ours)** | ✓ | ✓ | ✓ | ✓ |

variations, resulting in camera-specific models and absent knowledge transfer between these data silos. Therefore, such models ignore large amounts of data, limiting their robustness and cross-applicability. Furthermore, supervised downstream models are inherently limited by the availability of application-specific annotations. Given that manual labeling is time-intensive and often infeasible for large-scale datasets, self-supervised pre-training has emerged as a powerful alternative (He et al., 2022; Devlin et al., 2019; Caron et al., 2021; He et al., 2020). Empirical findings in Natural Language Processing have demonstrated that the effectiveness of self-supervised-learning (SSL) scales with the amount of training samples (Kaplan et al., 2020). This motivates the use of extensive cross-silo datasets to enhance pre-training. However, existing strategies are not camera-agnostic, restricting pre-training to camera-specific data silos and limiting their effectiveness. To overcome these obstacles, we propose a novel camera-agnostic model with a tailored SSL strategy that is capable of unlocking the data treasures of different cameras that are not yet accessible (Fig. 1). Our contribution is threefold:

1. **First approach to spatio-spectral camera-agnostic representation learning:** We propose the first method that enables spatio-spectral encoding in a camera-agnostic manner. To this end, we introduce wavelength positional encoding for establishing cross-camera channel correspondences, and learnable spectral representations for efficient representation learning.

2. **First camera-agnostic spatio-spectral self-supervision framework:** We propose a novel spectral feature-based SSL strategy tailored to CARL, which can be seamlessly combined with I-JEPA spatial pre-training (Assran et al., 2023) to form an end-to-end framework for camera-agnostic spatio-spectral self-supervised pre-training.

3. **Large-scale cross-domain validation:** We validated the proposed model in three application areas, specifically medical imaging, automotive vision, and satellite imaging. Across all experiments, our approach outperformed both camera-specific and channel-invariant baselines, demonstrating superior cross-modality knowledge transfer and unique robustness to spectral heterogeneity arising from simulated and real-world camera variations.

## 2 RELATED WORK

**Feature extraction strategies for spectral imaging** Generating rich image representations remains a fundamental challenge in computer vision, with significant implications for downstream tasks such as image segmentation. For 2D spectral images, the encoding process inherently spans three dimensions: two spatial dimensions and one spectral dimension. In datasets with uniform spectral properties, conventional models such as CNNs and Vision Transformers (ViTs) are commonly employed (Dosovitskiy et al., 2021; Theisen et al., 2024). However, these models focus solely on spatial encoding (2D projections, ViT blocks) and assume a fixed channel dimension. Recent approaches in remote sensing have introduced models that jointly encode spatial and spectral information, for example, by forming spatio-spectral patches (Cong et al., 2022; Hong et al., 2024). Yet, these methods lack invariance to the channel dimension, preventing their applicability to cross-camera datasets. Braham et al. (2024) addresses this by introducing a Spectral Adapter that resolves the channel dimension through 1D convolutions and pooling operations. However, it overlooks channel relationships derived from camera-specific wavelength information. To overcome this limitation, recent work has proposed channel-adaptive 2D projection layers that learn wavelength-conditioned projection matrices (Li et al., 2025a; Xiong et al., 2024; Varga et al., 2023; Wang et al., 2025). While effective, these methods rely on spatial operations and do not explicitly encode salient spectral information, which may reduce robustness on spectrally heterogeneous datasets. Alternative strategies encode pixel- or patch-wise

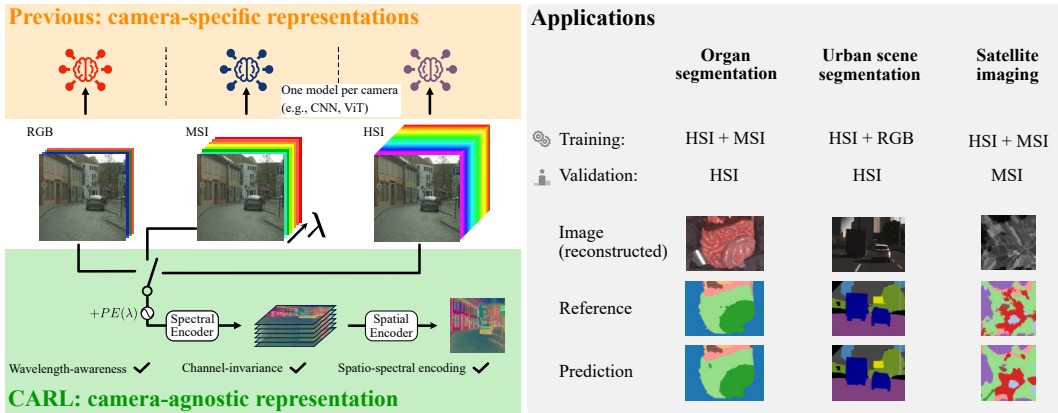

Figure 1: **CARL addresses spectral camera variations by learning camera-agnostic representations.** Unlike existing methods that require retraining for each channel configuration, CARL generalizes across cameras and outperforms both camera-specific and channel-invariant approaches across domains. The model processes one image at a time, ensuring flexibility without dependence on fusion strategies.

information only along the channel dimension (Hong et al., 2021; Hang et al., 2019; Seidlitz et al., 2022). However, these designs cannot model geometric structures, which significantly impairs downstream performance. Fusion-based methods offer another solution by aligning multi-modal data with varying channel dimensions through early, mid, or late fusion (Lin et al., 2023; Audebert et al., 2018). Typically, these architectures incorporate modality-specific projection layers or entire encoders before feeding into a multi-modal encoder (Astruc et al., 2025; Tseng et al., 2025; Jakubik et al., 2025; Fuller et al., 2023). Crucially, such methods assume access to all modalities during both training and inference. While this assumption may hold in remote sensing with standardized sensors, it is impractical for industrial or medical spectral imaging applications, where sensor diversity is broader, often unknown, and sensor-specific data silos contain relatively few samples.

**Self-supervised learning strategies for spectral imaging**  With growing compute resources and data availability, SSL strategies have gained importance in recent years. In RGB imaging, masked image modeling has emerged as a central paradigm, where input patches are randomly masked and reconstructed at the pixel level using a strong encoder paired with a lightweight decoder (He et al., 2022). This paradigm has been extended to spectral imaging, encompassing camera-specific spatio-spectral encoding (Hong et al., 2024; Cong et al., 2022) and camera-agnostic spatial modeling (Xiong et al., 2024; Wang et al., 2025; Sumbul et al., 2025). Building on advances in RGB vision, feature-based SSL has proven more efficient than pixel-based approaches (Caron et al., 2021; Assran et al., 2023), a benefit that is particularly important in spectral imaging, where pixel values are highly sensitive to factors such as atmospheric conditions in satellite data or illumination calibration in laboratory settings (Baumann et al., 2024). Accordingly, feature-based SSL has been adapted to spectral imaging models, though existing methods remain restricted to spatial encoding and spatial self-supervision (Tseng et al., 2025; Astruc et al., 2025; Waldmann et al., 2025). Notably, no SSL framework——whether camera-agnostic, feature-based, or both—has yet been designed to capture spatio-spectral encoding. An overview of existing approaches is provided in Tab. 1.

## 3 FRAMEWORK FOR CAMERA-AGNOSTIC SPECTRAL IMAGE ANALYSIS

In this paper, we present CARL, a novel model for spectral image processing, designed to unlock the potential of camera-specific data silos. As illustrated in Fig. 2, the proposed framework transforms camera-dependent spectral information into a camera-agnostic representation through a novel spectral encoder $E_{\text{spec}}$, followed by the extraction of geometric information through a standard spatial encoder $E_{\text{spat}}$. To establish cross-camera channel correspondences, we translate the concept of positional encoding, traditionally used for discrete token positions within transformers (Vaswani et al., 2017), to channel-specific wavelengths. To facilitate efficient representation learning along

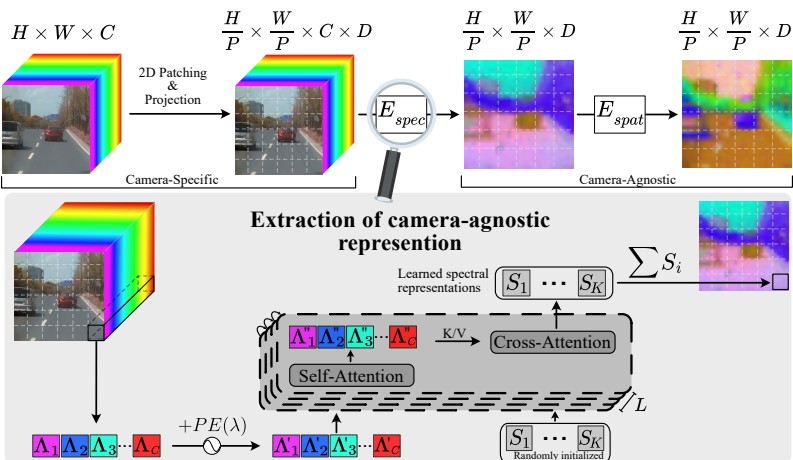

Figure 2: **Conversion of a camera-specific spectral image into a camera-agnostic representation.** To address the heterogeneity in camera-dependent spectral properties, a dedicated spectral encoder extracts a camera-agnostic representation by leveraging spectral tokens encoding wavelength information. A spectral image of dimension $H \times W \times C$ is divided into patches of size $P$ and projected band-wise into a $D$-dimensional feature space. The spectral encoder $E_{\text{spec}}$ processes each patch individually, and hereby resolves the spectral dimension. In particular, $E_{\text{spec}}$ encodes the wavelength $\lambda_i$ of channel $i$ as positional encoding $PE(\lambda_i)$ and adds it to the embedded patch $\Lambda_i$. Self-Attention across spectral tokens $(\Lambda_i)_{i \leq C}$ and Cross-Attention with $K$ learned spectral representations yield enriched representations $(S_j)_{j \leq K}$. After aggregation into a camera-agnostic representation, a standard image encoder, $E_{\text{spat}}$, captures spatial relationships.

the spectral dimension, we propose a novel encoder that distills channel information into a sparse set of spectral representations. Ultimately, the encoder produces a camera-agnostic feature map enriched with spectral attributes, which can be seamlessly forwarded to established transformer-based spatial encoders (e.g., ViT). The spatial encoder $E_{\text{spat}}$ operates subsequent to $E_{\text{spec}}$ and enhances feature representation by capturing spatial relationships. Two learning paradigms are proposed for optimizing the spectral representations derived in $E_{\text{spec}}$. Specifically, they are either learned implicitly by minimizing a downstream task-specific loss or explicitly by minimizing a self-supervised loss (CARL-SSL, as described in Sec. 3.2 and Fig. 3).

## 3.1 ARCHITECTURE OF CARL

Given a spectral image $I \in \mathbb{R}^{H \times W \times C}$ with arbitrary channel dimension, $C$, the objective is to derive a camera-agnostic representation that contains salient spectral information. To project $I$ to feature space, each channel is processed by a shared 2D convolution with kernel size and stride equal to the patch size $P$ and output channels $D$, yielding a tensor of dimensionality $\frac{H}{P} \times \frac{W}{P} \times C \times D$. A patch, denoted as $\Lambda = (\Lambda_1, \dots, \Lambda_C) \in \mathbb{R}^{C \times D}$, is then independently processed by the spectral encoder, $E_{\text{spec}}$, to generate a camera-agnostic representation. We first construct a positional encoding $PE(\lambda) \in \mathbb{R}^{C \times D}$ where $\lambda = (\lambda_1, ..., \lambda_C)$ and $\lambda_i$ corresponds to the wavelength of channel $i$ such that the model is capable of establishing channel correspondences across cameras with different wavelength specifications. In this work, we employ the sinusoidal Fourier Features (Tancik et al., 2020) to encode positional information within the spectral dimension, defined as:

$$PE(\lambda_i) = \left[\cos\left(2\pi\alpha\lambda_i B\right), \, \sin\left(2\pi\alpha\lambda_i B\right)\right]^T \in \mathbb{R}^D \tag{1}$$

where $\alpha \in \mathbb{R}$ is a scaling factor, $B \sim \mathcal{N}\left(0, \sigma^2\mathbf{I}\right) \in \mathbb{R}^{D/2}$, and $\sigma \in \mathbb{R}$. Here, both $\alpha$ and $\sigma$ are hyperparameters. Subsequently, $PE(\lambda)$ is added to the patch $\Lambda$, thereby encoding the position of each $\Lambda_i$ along the wavelength axis. As illustrated in Fig. 2, a self-attention-cross-attention module is introduced to process the spectral tokens, $(\Lambda_i)_{i \leq C}$, and derive spectral representations. Specifically, $K$ learnable $D$-dimensional spectral representations, denoted as $(S_j)_{j \leq K}$, are initialized from a truncated normal distribution. Following a self-attention block applied to the spectral tokens, the spectral representations, $(S_j)_{j \leq K}$, attend to the spectral tokens, $(\Lambda_i)_{i \leq C}$, via cross-attention,

effectively distilling the most salient information. This self-attention-cross-attention module is iterated $L$ times to learn enriched spectral representations, $(S_j)_{j \leq K}$. Subsequently, a readout function, in this instance summation, is applied to $(S_j)_{j \leq K}$ to aggregate the information into a camera-agnostic representation for the patch $\Lambda$. As the spectral encoder generates such a representation for each patch independently, the incorporation of spatial relationships necessitates the utilization of a subsequent spatial encoder. It is noteworthy that since $E_{\text{spec}}$ has encoded device-dependent spectral properties within the feature space, most common transformer-based spatial encoders, such as ViT, may be employed for spatial encoding. To ensure dimensional compatibility between the spectral and spatial encoder, layer normalization and a linear transformation are applied prior to spatial encoding. After capturing inter-patch relationships, a task-specific head can be added for the intended downstream application.

## 3.2 Self-supervised training strategy

Tailored to CARL, we propose a self-supervised pre-training strategy, CARL-SSL, to leverage large-scale unlabeled datasets. As illustrated in Fig. 3, the procedure is disentangled into spectral and spatial self-supervised pre-training within an end-to-end framework. While I-JEPA (Assran et al., 2023) is adapted for spatial self-supervision, we introduce a novel feature-based spectral SSL strategy. Given student encoders $E_{\text{spec}}$ and $E_{\text{spat}}$ from Sec. 3.1 along with their teacher counterparts, $\tilde{E}_{\text{spec}}$ and $\tilde{E}_{\text{spat}}$, which are updated via exponential moving average, we apply a masking strategy to specific regions of the students' input. The remaining tokens are then encoded by the student networks. The SSL objective is to predict the masked features generated by the teacher encoders, using dedicated predictors $\phi_{\text{spec}}$ and $\phi_{\text{spat}}$. Specifically, for an image $I \in \mathbb{R}^{H \times W \times C}$, the initial convolution is applied as described in Sec. 3.1. A spectral mask, denoted by $M \subseteq \{1, \ldots, C\}$, containing the masked channel indices, is sampled for a patch $\Lambda$, and the unmasked tokens, denoted by $(\Lambda_i)_{i \notin M}$, are forwarded to the student spectral encoder, $E_{\text{spec}}$, to generate spectral representations, $(S_j)_{j \leq K}$ (see Fig. 3). Conversely, the teacher spectral encoder receives all spectral tokens as input, producing learned spectral tokens, $(\tilde{\Lambda}_i)_{i \leq C}$, via self-attention, and spectral representations, $(\tilde{S}_j)_{j \leq K}$. The objective of spectral pre-training is then to predict the masked spectral tokens, $(\tilde{\Lambda}_i)_{i \in M}$, based on the student spectral representations, $(S_j)_{j \leq K}$, and the positional encoding of the masked wavelengths. To this end, a transformer-based predictor, denoted by $\phi_{\text{spec}}$, is employed, receiving as input a sequence with the spectral representations and dedicated mask tokens (Devlin et al., 2019). The mask tokens are $|M|$ copies of a shared, learnable embedding and are summed with wavelength positional encoding $(PE(\lambda_i))_{i \in M}$, corresponding to the masked wavelengths.

Subsequently, $\phi_{\text{spec}}$ processes this sequence through multiple self-attention blocks, resulting in learned mask tokens as predictions for $(\tilde{\Lambda}_i)_{i \in M}$. Network optimization uses the VICReg loss (Bardes et al., 2021) on the spectral predictions, denoted as $\mathcal{L}_{\text{spec}}$, which comprises invariance, variance, and covariance terms. The invariance term minimizes the mean-squared error between predicted and target spectral tokens, while the variance and covariance terms contribute to training stability and the prevention of feature collapse. For joint spatial training, the spectral representations from both the student and teacher encoders are aggregated into 2D camera-agnostic representations. Subsequently, a 2D region of the student's feature representation is masked, and the remaining spatial tokens are processed by $E_{\text{spat}}$. Analogous to spectral pre-training, the spatial predictor, $\phi_{\text{spat}}$, receives the student features and the positional encoding of the masked tokens as input, and predicts the masked features generated by the teacher spatial encoder, $\tilde{E}_{\text{spat}}$. The spatial loss function, $\mathcal{L}_{\text{spat}}$, is likewise defined using the VICReg loss on the spatial predictions. Finally, the overall training objective is

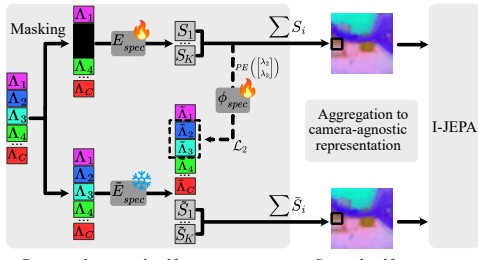

*Proposed spectral self-supervision*     *Spatial self-supervision*

Figure 3: **CARL-SSL enables joint learning of camera-agnostic representations and spatial relations.** Spectral self-supervision involves reconstruction of masked spectral channels in feature space. The student $E_{\text{spec}}$ extracts spectral representations $(S_j)_{j \leq K}$ from a spectrally masked input, while the predictor $\phi_{\text{spec}}$ predicts the masked spectral tokens using masked wavelengths , and $(S_j)_{j \leq K}$. Target tokens are generated by the teacher $\tilde{E}_{\text{spec}}$ from the complete input. The aggregated camera-agnostic representations are subsequently processed by I-JEPA.

Table 2: **CARL-SSL demonstrates superior performance compared with both camera-specific and camera-agnostic models.** The mIoU scores with the 95 % confidence intervals on the HSICity test set. While the camera-specific model was pre-trained on Cityscapes and fine-tuned exclusively on HSICity, the other models are channel-invariant adaptations which were concurrently trained on both datasets.

| | Camera-specific model | Spectral Adapter | HyperFree | Hyve | DOFA | CARL | CARL-SSL |
|---|---|---|---|---|---|---|---|
| mIoU | 44.6 [40.9; 47.3] | 43.4 [41.0; 45.2] | 44.6 [42.2; 46.5] | 48.0 [45.4; 50.0] | 49.6 [46.8; 51.6] | 48.6 [45.6; 51.0] | **50.1** [47.2; 52.4] |

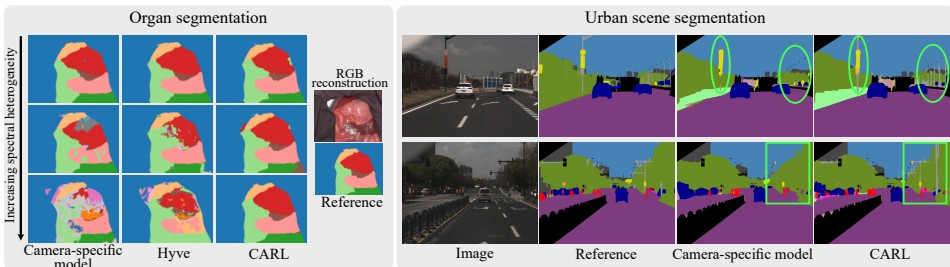

Figure 4: **Our model enables cross-modality knowledge transfer. (Left)** With increasing spectral heterogeneity in the training set, both the camera-specific model and Hyve exhibit a notable rise in prediction noise. In contrast, CARL consistently provides accurate predictions. **(Right)** In two HSICity test set examples, the HSI-specific model fails to segment "poles" (gray labels) due to their absence in the HSICity training set. CARL, however, jointly trained on RGB and HSI data, effectively leverages "pole" annotations from Cityscapes to inform its predictions on HSICity.

given by $\mathcal{L} = \mathcal{L}_{\text{spat}} + \mathcal{L}_{\text{spec}}$ to jointly optimize the student encoders and predictors. Further details are provided in appendix A.2.

### 3.3 IMPLEMENTATION DETAILS

Following ablation studies (see Sec. 5), we set $\sigma = 3$, as defined in Eq. (1) for the wavelength positional encoding, and the number of spectral representations within the spectral encoder to $K = 8$. Importantly, these hyperparameters are held constant across all three application domains, underscoring their generality. Unless otherwise stated in the experiments, we employed EVA-02 (Fang et al., 2024) as spatial encoder, which is a modern version of ViT (Dosovitskiy et al., 2021). Further implementation details of the model are outlined in appendix A.

## 4 EXPERIMENTS AND RESULTS

The experiments aimed to address the following research questions pertaining to the proposed model:

(RQ1) In silico proof of concept: Does the spectral representation learning approach enable effective knowledge transfer across cameras?

(RQ2) Real-world cross-domain generalization: To what extent do the proposed model and the spectral self-supervision approach help in handling real-world cross-camera variations?

Classification performance is reported as overall accuracy (OA), whereas segmentation performance is measured by intersection-over-union (IoU).

### 4.1 IN SILICO PROOF OF CONCEPT: MEDICAL IMAGING

**Datasets.** Synthetic multispectral images were generated from a real hyperspectral dataset, enabling isolated control over spectral variations while preserving spatial context. We employed a private collection of porcine organ images, semantically annotated into 19 classes (Seidlitz et al., 2022).

Table 3: **CARL excels at cross-sensor learning across different application domains.** **(a)** Hyperspectral images per subject (S) transformed to multispectral images using real-world filters (F). CARL achieves stable performance in contrast to existing methods. **(b)** CARL exhibits the small drop in IoU when traffic light and sign classes were removed from HSI training set. **(c)** Ablation study on the loss components of CARL-SSL. Overall accuracy (OA) is reported on the m-forestnet validation set using a feature-based $k$-NN classifier.

(a) Medical Data

| Training Data | Hyve | DOFA | CARL |
|---|---|---|---|
| Synthetic MSI (4S, 1F) Real HSI (8S) | 52.1 | 58.1 | **64.6** |
| Synthetic MSI (4S, 2F) Real HSI (8S) | 47.7 | 49.2 | **60.3** |
| Synthetic MSI (6S, 1F) Real HSI (6S) | 35.4 | 52.0 | **62.1** |

(b) Automotive Data

| Class | Hyve | DOFA | CARL |
|---|---|---|---|
| Traffic Light ($\Delta$) | 15.5 $-37.5$ | 8.4 $-50.4$ | **29.2** $-24.5$ |
| Traffic Sign ($\Delta$) | 10.9 $-43.8$ | 14.9 $-45.0$ | **31.7** $-26.7$ |
| mIoU ($\Delta$) | 42.7 $-5.3$ | 42.7 $-6.9$ | **46.2** $-2.4$ |

(c) Ablation on CARL-SSL

| SSL Strategy | OA |
|---|---|
| Spatial SSL $\mathcal{L}_{\text{spat}}$ + Spectral SSL $\mathcal{L}_{\text{spec}}$ | 22.1 **32.6** |
| = CARL-SSL | |

Acquisition was performed with a Tivita® Tissue HSI camera (Diaspective Vision GmbH, Am Salzhaff, Germany), capturing 100 spectral channels spanning 500 nm to 1,000 nm. The training set comprises 254 images from 12 subjects. To emulate realistic multispectral acquisition with optical filters, we modeled each filter response as a Gaussian function. Specifically, the number of channels $C$ in a virtual multispectral camera was first sampled uniformly from $\{10, \dots, 25\}$. The corresponding $C$ center wavelengths, serving as the Gaussian means, were selected within 550 nm to 950 nm using farthest point sampling. To obtain a realistic range of variances, we fitted Gaussian functions to the filter responses of a commercial multispectral camera and sampled $C$ variance values from this range. Given the sampled means and variances of $C$ channels, the filter functions were constructed and applied to an hyperspectral image to generate the corresponding multispectral image via matrix multiplication. In this way, we simulated six distinct camera configurations and progressively replaced hyperspectral images in the training set with their multispectral counterparts on a per-subject basis, while keeping the hyperspectral validation and test sets unchanged. This protocol isolates spectral variability and allows for a rigorous assessment of model robustness to spectral heterogeneity. Additional details on the data generation can be found in appendix F.

**Baseline methods.** The model's performance was benchmarked against a camera-specific baseline and six channel-invariant methods, which can be grouped into three categories: spatio-spectral encoding (Spectral Adapter (Braham et al., 2024)); channel-adaptive embedding layers (DOFA (Xiong et al., 2024), Hyve (Varga et al., 2023), and HyperFree (Li et al., 2025a)); and camera-specific embedding layers (Early Fusion (Astruc et al., 2025)). The camera-specific model employs a standard U-Net, representing the state of the art on the original hyperspectral dataset (Seidlitz et al., 2022), and was trained exclusively on the hyperspectral subset of each training set variant. The other methods were integrated with either a U-Net or a ViT-based architecture, depending on which proved most compatible.

For ViT-based methods, including CARL, we adopted the ViT-Adapter (Chen et al., 2023) for hierarchical features and UperNet (Xiao et al., 2018) for segmentation. All models followed the same training protocol.

**Results.** The mIoU scores as a function of the fraction of multispectral subjects within the training set is presented in Fig. 5. The proposed method uniquely maintained a high mIoU across all training set variants. This is qualitatively confirmed in Fig. 4, where prediction noise of the baseline methods increases with spectral heterogeneity, while our model remains stable and accurate. While simulated filter functions enable scalable evaluation under spectral heterogeneity, realistic modeling remains crucial. We therefore conducted an additional experiment in which multispectral images were synthesized using real-world filters instead of Gaussian approximations. As shown in Tab. 3a, CARL consistently outperforms baseline methods across all training data configurations.

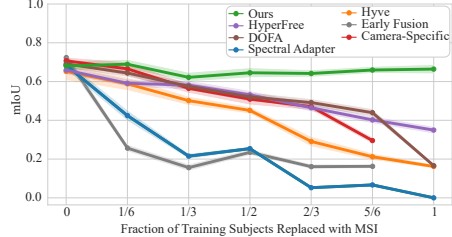

Figure 5: **Our model shows unique robustness to spectral heterogeneity in the organ experiments.** As spectral heterogeneity increases with the multispectral replacements in the training set, CARL uniquely maintains a high mIoU score on the hyperspectral test set. Shaded area: 95 % confidence interval.

Table 4: **CARL learns strong in-distribution features.** Linear-probing results on four Sentinel-2 benchmarks. CARL attains the highest accuracy on three of the four datasets and the best average rank across all eleven benchmark datasets.

| | m-bigearthnet | m-eurosat | m-cashew | m-SA-crop-type | Rank over 11 datasets |
|---|---|---|---|---|---|
| SpectralGPT$^+$ | 45.0 | 69.9 | 14.5 | 13.7 | 5.5 |
| Galileo | 49.8 | 84.2 | 10.5 | 19.3 | 5.5 |
| Croma | 59.5 | 86.6 | 12.1 | 25.2 | 5.0 |
| DOFA | 61.0 | 89.9 | 18.2 | 21.7 | 3.2 |
| Copernicus-FM | 62.1 | 87.2 | 14.5 | **26.5** | 2.6 |
| SMARTIES | 62.0 | **92.6** | 12.7 | 24.3 | 2.6 |
| CARL | **69.0** | 84.4 | **18.9** | 26.5 | **1.6** |

Table 5: **CARL produces robust features on unseen sensors.** Linear-probing results on four out-of-distribution sensors. Despite large spectral heterogeneity, CARL achieves the best performance on three of the four datasets, demonstrating strong cross-sensor generalization.

| | LoveDA Urban | m-forestnet | WHU-OHS | Wuhan |
|---|---|---|---|---|
| Sensor | RGB (3 bands) | LandSat-8 (6 bands) | Orbita (32 bands) | Gaofen-5 (116 bands) |
| DOFA | 12.6 | 43.8 | 1.5 | 20.3 |
| Copernicus-FM | 15.4 | 44.8 | 1.5 | 18.1 |
| SMARTIES | 13.5 | **49.8** | 1.5 | 18.8 |
| CARL | **29.0** | 47.0 | **21.7** | **21.5** |

## 4.2 REAL-WORLD EVALUATION: AUTOMOTIVE

The purpose of this experiment was to test the capabilities of our approach under real-world conditions in the context of autonomous driving. Specifically, we investigated whether our model can effectively leverage RGB and hyperspectral images for urban scene segmentation.

**Datasets.** The Cityscapes RGB dataset (Cordts et al., 2016), comprising semantic annotations for 19 classes, was employed alongside its hyperspectral counterpart, HSICity (Shen et al.). HSICity contains images with 128 channels (450 nm to 950 nm) and shares the same labels. However, its training set suffers from coarse annotations and class imbalance; for instance, the "pole" class is present in the test set but absent from the training set.   We therefore added finely annotated Cityscapes images containing "pole" labels, resulting in 4,029 training images (1,054 from HSICity). To assess cross-modality learning, models were trained on this combined dataset and evaluated on the HSICity test set. For self-supervised pre-training, we leveraged a collection of heterogeneous urban datasets, including Cityscapes, HSICity, and the multispectral datasets HyKo-VIS (Winkens et al., 2017) and HSIDrive (Basterretxea et al., 2021).

**Baseline methods.** A Swin Transformer (Liu et al., 2021) with a Mask2Former head (Cheng et al., 2022), referred to as SwinMask2Former, pre-trained on the Cityscapes dataset, was adapted through the integration of a channel-invariant module. Our spectral encoder, serving as this module, was compared with channel-adaptive layers of HyperFree, DOFA, and Hyve, as well as with the Spectral Adapter. Additionally, as a camera-specific baseline, the RGB-pre-trained SwinMask2Former was trained exclusively on HSICity. All models adhered to an identical training protocol.

**Results.** The mIoU scores on the HSICity test set are presented in Tab. 2. CARL-SSL demonstrated superior performance compared to the baseline methods. Due to the absence of the "pole" class in the HSICity training set, the camera-specific model failed to segment any poles in the test set, despite RGB-pre-training (see Fig. 4). In contrast, our model effectively transferred knowledge from the "pole" labels in Cityscapes to improve its predictions on HSICity, achieving the highest IoU for this class. To further assess cross-modality learning, we removed the "traffic light" and "traffic sign" classes from the HSICity training set, and re-trained the models. As shown in Tab. 3b, CARL most effectively leveraged RGB supervision to achieve superior HSI predictions for the excluded classes on the test set compared to the baselines.

Table 6: **Ablation studies on CARL.** **(a)** Wavelength positional encoding (PE) is essential, with $\sigma = 3$ yielding the best performance. **(b)** Summation proves to be the most effective strategy for aggregating spectral representations, and **(c)** using $K = 8$ spectral representations suffices to distill the channels. **(d)** Moreover, CARL benefits from larger embedding dimensions.

(a) Positional Encoding

| Method | mIoU |
|---|---|
| No PE | 18.3 |
| PE ($\sigma = 1$) | 55.1 |
| PE ($\sigma = 3$) | **61.5** |
| PE ($\sigma = 10$) | 57.2 |

(b) Aggregation

| Method | mIoU |
|---|---|
| Summation | **62.7** |
| Concatenation | 61.8 |
| Maximum | 61.8 |
| Attention Pooling | 60.0 |

(c) # Spectral Rep.

| # Spectral Rep. | mIoU |
|---|---|
| $K = 1$ | 57.8 |
| $K = 4$ | 58.2 |
| $K = 8$ | **63.9** |
| $K = 16$ | 62.2 |

(d) Feature Dim.

| Size | mIoU |
|---|---|
| $D = 384$ | 64.4 |
| $D = 768$ | 66.2 |

## 4.3 REAL-WORLD EVALUATION: SATELLITE IMAGING

The third experiment aimed to evaluate the capabilities of CARL-SSL in satellite imaging.
**Dataset & Baseline methods.** To facilitate benchmarking against strong pre-trained baselines such as SpectralGPT[+] and DOFA, we scaled up pre-training to a corpus of approximately 800,000 images, comprising Sentinel-2 multispectral data (Sumbul et al., 2019) and EnMAP hyperspectral data (Fuchs and Demir, 2023; Braham et al., 2024). Feature quality was assessed via linear probing on eleven datasets (Lacoste et al., 2023; Li et al., 2022; Wang et al., 2021; Hong et al., 2023). These include five in-distribution datasets (Sentinel-2) and six datasets captured by sensors not present in the pre-training set. Further dataset details are given in Tabs. 7, 8. We compared against six remote-sensing foundation models, three of which are camera-agnostic and therefore can be evaluated on every dataset.
**Results.** Results are summarized in Tabs. 4, 5, 9. CARL delivers consistently strong performance across the benchmarks and achieves the best average rank across all eleven datasets. In particular, CARL exhibits robust generalization to OOD sensors, outperforming the second-best method by a substantial margin on several datasets. We attribute this advantage to CARL's camera-agnostic spatio-spectral encoding scheme, which is unique in comparison to the baseline methods.

## 5 ABLATION STUDY

The ablation study in Tab. 6 examines the contribution of CARL's key architectural components. Training was conducted on a dataset variant from Sec. 4.1, while evaluation was performed on an HSI validation split. Removing the wavelength positional encoding severely impairs the model's ability to align channels across different cameras. For aggregating the spectral representations, simple summation achieves the highest accuracy while remaining computationally efficient. With respect to the number of spectral representations, we find that $K = 8$ tokens are sufficient to capture the essential spectral information. Notably, performance gains beyond this point are primarily achieved by increasing the embedding dimension rather than $K$. We also conducted an analysis of CARL-SSL's two self-supervision tasks—spectral and spatial—as summarized in Tab. 3c. To isolate their individual contributions, we performed two pre-training variants under a reduced training budget: one employing only spatial self-supervision, and the other utilizing our proposed spatio-spectral strategy. The resulting image embeddings were evaluated using a $k$-NN classifier on the m-forestnet validation set (Lacoste et al., 2023). The model trained with spatial self-supervision alone exhibited a collapse in the spectral representations, leading to significantly reduced accuracy. In contrast, incorporating spectral self-supervision effectively mitigated this collapse and yielded substantially stronger representations, resulting in a $+10.5$ OA improvement.

## 6 DISCUSSION

We introduced CARL, to our knowledge the first camera-agnostic framework that unifies spatio-spectral encoding with spatio-spectral SSL pre-training. Our approach tackles a critical gap in spectral image processing: the lack of a representation learning framework that generalizes across spectrally heterogeneous datasets. We demonstrated its effectiveness in both traditional satellite imaging and in domains such as medical imaging, where sensor variability is particularly pronounced due to the diversity of commercial camera manufacturers. Adaptive embedding approaches, including HyperFree, Hyve, and DOFA, rely solely on spatial operations and neglect crucial inter-channel

relationships, resulting in limited performance. Alternative models, such as SpectralGPT$^+$, introduce spatio-spectral encoding, but depend on fixed channel dimensions, preventing generalization across different spectral cameras. In contrast, CARL employs wavelength-aware spatio-spectral encoding that is independent of channel dimensionality, enabling robust generalization under spectral heterogeneity and scalability across modalities, as demonstrated in both pre-training (Tabs. 2, 4, 5, 9) and downstream tasks (Figs. 4, 5, Tab. 3). Limitations of our work include higher computational cost compared to channel-adaptive embedding approaches (see appendix D), as well as challenges from sensor heterogeneity beyond spectral properties, such as differences in spatial resolution. The latter may be mitigated by incorporating additional sensor metadata. Despite these limitations, our approach successfully integrates real-world cross-camera datasets and outperforms existing methods. By unlocking the untapped potential of cross-sensor spectral datasets, CARL paves the way toward a more universal and accessible future in spectral imaging.

## 7 REPRODUCIBILITY STATEMENT

To ensure reproducibility, we have provided a detailed description of our method in Sec. 3, with additional implementation details in appendix A. Pseudo-code illustrating the forward passes of CARL and CARL-SSL can be found in algorithms 1, 2. Furthermore, all datasets used in Secs. 4.2, 4.3 are publicly available. Finally, full per-class IoU scores are reported in Tabs. 10, 12.

## 8 ACKNOWLEDGMENTS

This project was supported by the European Research Council (ERC) under the European Union's Horizon 2020 research and innovation program (NEURAL SPICING, 101002198), the National Center for Tumor Diseases (NCT), Heidelberg's Surgical Oncology Program, the German Cancer Research Center (DKFZ), and the Helmholtz Association under the joint research school HIDSS4Health (Helmholtz Information and Data Science School for Health). We also acknowledge the support through state funds for the Innovation Campus Health + Life Science Alliance Heidelberg Mannheim from the structured postdoc program for Alexander Studier-Fischer: Artificial Intelligence in Health (AIH).

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

# Appendix

## A   IMPLEMENTATION DETAILS

We begin by outlining general implementation details of CARL. In the experiments from Secs. 4.1, 4.2, we employed a small version of CARL, comprising overall 12 attention blocks with an embedding dimension of 384. For the satellite experiments in Sec. 4.3, we switched to the base version of CARL, keeping the same 12-block depth but increasing the embedding dimension of the spatial encoder to 768 to align with SpectralGPT$^+$ and DOFA. The blocks were structured into $L = 4$ self-attention-cross-attention modules within the spectral encoder and 8 self-attention blocks within the spatial encoder. To incorporate wavelength information in the positional encoding, the wavelengths were given in nanometers, and scaled by a factor of $\alpha = 10^{-3}$, to obtain position coordinates approximately in the range of one. In accordance with the ablation studies presented in the main manuscript, the hyperparameter $\sigma$ in the wavelength positional encoding was set to 3 and $K = 8$ spectral representations were employed. The initial convolution in our model utilized a kernel size and stride of 8, resulting in patch dimensions of $8 \times 8$. The spectral representations within the spectral encoder were implemented as learnable embeddings, initialized using a truncated normal distribution with a mean of zero and a standard deviation of 0.5. Furthermore, a 1D sinusoidal positional encoding scheme, based on discrete token positions, was implemented to represent the positions of $(S_j)_{j \leq K}$. Pseudo-code of a forward step of CARL can be found in algorithm 1.

### A.1   IMPLEMENTATION DETAILS OF DOWNSTREAM TASKS

In the first and third experiment, CARL was integrated with the ViTAdapter (Chen et al., 2023) to generate hierarchical features, which were subsequently forwarded to the UperNet segmentation head (Xiao et al., 2018). The ViTAdapter applies lightweight convolutions to the input and facilitates information exchange with the spatial transformer via injector and extractor modules. To maintain invariance with respect to the channel dimensionality, a single channel of the given spectral image was used as input to these convolutions. The rationale behind this approach is to leverage the enhanced spatial resolution provided by the convolutions, as the encoded spectral information is contained within the camera-agnostic representation of the proposed model. As segmentation loss, an equally weighted sum of the cross entropy loss and the dice loss was employed. To enhance training stability, the attention blocks within the spectral encoder were initialized using Dinov2 weights (Oquab et al., 2023), which provide a robust checkpoint derived from self-supervised training on a large-scale dataset. The spatial encoder, EVA-02 (Fang et al., 2024), was initialized with self-supervised weights obtained through masked image modeling (He et al., 2022) on ImageNet (Deng et al., 2009). Furthermore, the model was optimized using the AdamW optimizer with an initial learning rate of $10^{-4}$. An exponential learning rate scheduler was employed to reduce the learning rate throughout the training process. In the second experiment, the training procedure followed that of the original Mask2Former (Cheng et al., 2022) training on Cityscapes (Cordts et al., 2016). In all experiments, a channel sampling strategy was employed to reduce the GPU memory consumption during training. Specifically, in instances where the channel dimension of the spectral image surpassed 32, a random subsampling of 32 channels was performed. Crucially, this did not affect performance, as the spectral tokens are ordered via wavelength positional encoding. As a result, all experiments were successfully executed on a single GPU endowed with a memory capacity of up to 40GB.

### A.2   IMPLEMENTATION DETAILS OF CARL-SSL

The model subject to pre-training consisted of the proposed spectral encoder, in conjunction with the EVA-02 spatial encoder. As outlined in the main manuscript, the pre-training strategy employs spectral and spatial masking with reconstruction objectives in the feature space through predictor networks. Pseudo-code of CARL-SSL is outlined in algorithm 2. To manage compute resources, we downsampled each hyperspectral image to 64 channels. For spectral self-supervision on hyperspectral EnMAP data, we applied a single mask covering 15 % to 30 % of the channels; for Sentinel-2 (multispectral) we masked two to three channels. Our spatial SSL strategy used two masks, each spanning 30 % to 50 % of the image area. The predictors are transformer architectures with a depth

---

**Algorithm 1:** Pseudo-code of a forward pass of CARL.

```
# Input:
# x: batch of images of shape (B, C, H, W)
# w: wavelengths corresponding of shape (B, C)

def forward(self, x, w):
    # --------------------- Projection -----------------------
    x = rearrange(x, "B C H W -> (B C) 1 H W")
    x = self.projection(x) # shape: (B*C, D, h, w)
    x = rearrange(x, "(B C) D h w -> (B h w) C D")
    # -------------- Spectral Positional Encoding --------------
    spec_pe = self.spec_pe(w) # shape: (B, C, D)
    spec_pe = repeat(spec_pe, "B C D -> (B h w) C D")
    x = x + spec_pe
    spec_reps = repeat(self.spec_reps, "1 K D -> (B h w) K D")
    spec_rep_pe = repeat(self.spec_rep_pe, "1 K D -> (B h w) K D")
    spec_reps = spec_reps + spec_rep_pe
    # -------------------- Spectral Attention -------------------
    for self_attn, cross_attn in self.spectral_blks:
        x = self_attn(x)
        spec_reps = cross_attn(spec_reps, x, x)
    # ------------- Spectral-to-Spatial Transition -------------
    spec_rep = spec_reps.sum(dim=1) # shape: (B*h*w, D)
    spec_rep = self.linear(self.norm(spec_rep))
    spec_rep = rearrange(spec_rep, "(B h w) D -> B (h w) D")
    # -------------- Spatial Positional Encoding ---------------
    spat_pe = self.spat_pe(h, w) # shape: (1, h*w, D)
    spat_pe = repeat(spat_pe, "1 N D -> B N D")
    spec_rep = spec_rep + spat_pe
    # -------------------- Spatial Attention --------------------
    for self_attn in self.spatial_blks:
        spec_rep = self_attn(spec_rep)
    spec_rep = self.norm(spec_rep) # shape: (B, h*w, D)
    # --------------------- Output Head ---------------------
    out = self.head(spec_rep)
    return out
```

---

of 3 and an embedding dimension of 384. Both, spectral and spatial self-supervision leverage the VICReg loss (Bardes et al., 2021), which is decomposed into invariance, variance, and covariance terms. Following the original recommendations, weights of 1 were assigned to the invariance and variance terms, and a weight of 0.05 was assigned to the covariance term. Let $\hat{y}$ and $y$ be the predicted and target tokens, both of shape $(B, N, D)$—with $B$ as batch size, $N$ sequence length, and $D$ embedding dimension. We then compute:

$$\mathcal{L} = \mathcal{L}_{\text{spec}} + \mathcal{L}_{\text{spat}} \tag{2}$$

$$= \mathcal{L}_{\text{vicreg}}(\hat{y}_{\text{spec}}, y_{\text{spec}}) + \mathcal{L}_{\text{vicreg}}(\hat{y}_{\text{spat}}, y_{\text{spat}}) \tag{3}$$

$$\mathcal{L}_{\text{vicreg}}(\hat{y}, y) = \mathcal{L}_{\text{inv}}(\hat{y}, y) + \mathcal{L}_{\text{var}}(\hat{y}) + 0.05 \cdot \mathcal{L}_{\text{cov}}(\hat{y}) \tag{4}$$

$$\mathcal{L}_{\text{inv}}(\hat{y}, y) = \frac{1}{B \cdot N \cdot D} \sum_{b=1}^{B} \sum_{n=1}^{N} \sum_{d=1}^{D} (\hat{y}_{b,n,d} - y_{b,n,d})^2 \tag{5}$$

$$\mathcal{L}_{\text{var}}(\hat{y}) = \frac{1}{N \cdot D} \sum_{n=1}^{N} \sum_{d=1}^{D} \max\left(0, 1 - \sqrt{\text{Var}(\hat{y}_{n,d})}\right) \tag{6}$$

$$\mathcal{L}_{\text{cov}}(\hat{y}) = \frac{1}{N \cdot D} \sum_{n=1}^{N} \sum_{\substack{i,j \le D \\ i \ne j}} \text{Cov}(\hat{y}_n)_{i,j}^2 \tag{7}$$

$$\text{Cov}(\hat{y}_n) = \frac{1}{B-1} \sum_{b=1}^{B} (\hat{y}_{b,n} - \overline{\hat{y}_n})(\hat{y}_{b,n} - \overline{\hat{y}_n})^T \quad \text{where } \overline{\hat{y}_n} = \frac{1}{B} \sum_{b=1}^{B} \hat{y}_{b,n} \tag{8}$$

While $\mathcal{L}_{\text{inv}}$ encourages $\hat{y}$ to resemble $y$, $\mathcal{L}_{\text{var}}$ promotes diversity among generated tokens by increasing their variance across the samples, preventing feature collapse. Finally, the covariance term minimizes off-diagonal absolute values in the feature covariance matrix, encouraging independence between feature components. This reduces redundancy and increases information across the feature dimension.

### A.3 CARL-SSL ON SATELLITE IMAGES

The pre-training dataset consisted of three distinct datasets, specifically HySpecNet-11k (Fuchs and Demir, 2023), SpectralEarth (Braham et al., 2024), and BigEarthNet (Sumbul et al., 2019), whose detailed composition is summarized in Tab. 7. The student and predictor networks were optimized

Table 7: **Composition of our remote sensing SSL-data.** Three datasets were used for self-supervised pre-training on satellite images. The table reports the number of images, sensor, number of spectral channels, and covered wavelength range. [1]Only a subset of the full SpectralEarth dataset was used in our pre-training.

| Dataset | # Images | Sensor | # Channels | Wavelength Range |
|---|---|---|---|---|
| HySpecNet-11k | 11,483 | EnMap | 202 | 418 nm to 2,445 nm |
| SpectralEarth | 247,030[1] | EnMap | 202 | 418 nm to 2,445 nm |
| BigEarthNet | 549,488 | Sentinel-2 | 12 | 443 nm to 2,202 nm |

**Algorithm 2:** Pseudo-code of a training step of CARL-SSL.

```
# Inputs:
# x: batch of images of shape (B, C, H, W)
# w: corresponding wavelengths of shape (B, C)

def training_step(x, w):
    # ---------------------------- Mask Sampling ----------------------------
    spec_enc_mask, spec_pred_mask = sample_spec_masks()
    spat_enc_mask, spat_pred_mask = sample_spat_masks()
    # ---------------------- Target Token Generation ----------------------
    with torch.no_grad():
        spec_teacher_tokens, spat_teacher_tokens = self.teacher(x, w)
        target_spec_tokens = apply_masks(spec_teacher_tokens, spec_pred_mask)
        target_spat_tokens = apply_masks(spat_teacher_tokens, spat_pred_mask)
    # --------------------- Student Token Generation ---------------------
    x = rearrange(x, "B C H W -> (B C) 1 H W")
    x = self.student.projection(x) # shape: (B*C, D, h, w)
    x = rearrange(x, "(B C) D h w -> (B h w) C D")
    # ------------------------ Spectral Masking ------------------------
    spec_student = apply_masks(x, spec_enc_mask)
    w_student = apply_masks(w, spec_enc_mask)
    # ------------------------ Spectral Encoding ------------------------
    student_spec_reps = self.student.spec_encoder(spec_student, w_student)
    # ------------------------ Spectral Prediction ------------------------
    w_teacher = apply_masks(w, spec_pred_mask)
    pred_spec_tokens = self.spec_predictor(student_spec_reps, w_teacher)
    # ------------------- Spectral-to-Spatial Transition -------------------
    student_spec_rep = student_spec_reps.sum(dim=1) # shape: (B*h*w, D)
    student_spec_rep = self.linear(self.norm(student_spec_rep))
    student_spec_rep = rearrange(student_spec_rep, "(B h w) D -> B (h w) D")
    # ------------------------ Spatial Masking ------------------------
    spat_student = apply_masks(student_spec_rep, spat_enc_mask)
    spat_student = self.student.spat_encoder(spat_student, spat_enc_mask)
    # ------------------------ Spatial Prediction ------------------------
    pred_spat_tokens = self.spat_predictor(spat_student, spat_pred_mask)
    # ------------------------ Loss Computation ------------------------
    loss_spec = VICREG(pred_spec_tokens, target_spec_tokens)
    loss_spat = VICREG(pred_spat_tokens, target_spat_tokens)
    loss = loss_spec + loss_spat
    # ------------------------ Optimization Step ------------------------
    self.optimizer.zero_grad()
    loss.backward()
    self.optimizer.step()
    # ------------------------ EMA Teacher Update ------------------------
    update_weights(self.teacher, self.student)
```

using AdamW with an initial learning rate of $10^{-4}$, weight decay of $0.04$, and a cosine annealing learning rate scheduler with a final learning rate of $10^{-6}$. Teacher networks were updated each iteration using an exponential moving average (EMA) of student weights, with momentum linearly increased from 0.996 to 1.0 during pre-training, following (Assran et al., 2023). Training ran for 78,000 iterations on an NVIDIA H200 GPU with a batch size of 64.

Linear-probe evaluations ran for 50 epochs on GeoBench datasets and 30 epochs on SegMunich, using Adam with a 0.001 starting learning rate and a cosine annealing schedule. We applied random resized cropping for augmentation. Because our model operates on $8 \times 8$ patches while DOFA uses $16 \times 16$, we doubled DOFA's input resolution to guarantee a fair comparison.

## B  EXPANDED RELATED WORK

In this section, we will elaborate on existing approaches to spectral image encoding and our specific contribution. To this end, we categorize related work, and discover gaps in the literature.

**Feature extraction for spectral imaging**  Modern RGB image encoders typically start with a 2D patch projection that maps image patches into a feature space, followed by Vision Transformer blocks

Table 8: **Details of benchmark datasets for remote sensing experiments.** The first block lists five in-distribution Sentinel-2 datasets that are present in CARL's pretraining set. The second block lists datasets captured by sensors that were unseen during pretraining, suitable for evaluating cross-sensor generalization.

| Dataset | Sensor / Platform | Channels | Wavelength (nm) | Classes | Task | Zero-shot |
|---|---|---|---|---|---|---|
| *In-distribution sensors (present in pretraining)* | | | | | | |
| SegMunich | Sentinel-2 | 10 | 442–2202 | 12 | Seg | ✗ |
| m-bigearthnet | Sentinel-2 | 12 | 442–2202 | 43 | Cls | ✗ |
| m-eurosat | Sentinel-2 | 13 | 442–2202 | 10 | Cls | ✗ |
| m-cashew | Sentinel-2 | 13 | 442–2202 | 7 | Seg | ✗ |
| m-SA-crop-type | Sentinel-2 | 13 | 442–2202 | 10 | Seg | ✗ |
| *Out-of-distribution senors (unseen during pretraining)* | | | | | | |
| LoveDA Urban | Google Earth | 3 | RGB | 6 | Seg | ✓ |
| LoveDA Rural | Google Earth | 3 | RGB | 6 | Seg | ✓ |
| m-forestnet | LandSat-8 | 6 | 482–2200 | 12 | Cls | ✓ |
| WHU-OHS | Orbita hyperspectral sat. | 32 | 466–940 | 24 | Seg | ✓ |
| Wuhan | Gaofen-5 | 116 | 420–2400 | 13 | Seg | ✓ |
| Beijing | Gaofen-5 | 116 | 420–2400 | 13 | Seg | ✓ |

or their variants. These architectures are designed for RGB inputs and therefore do not natively handle sensors with arbitrary numbers of spectral channels. Early adaptations for spectral imaging add multiple projection layers to accommodate different channel counts (Tseng et al., 2025; Astruc et al., 2025), or replace the single projection layer with complete modality-specific encoders (Jakubik et al., 2025; Fuller et al., 2023; Astruc et al., 2024). Such designs—often implemented as early- or mid-fusion models—work well for the sensors seen during training but cannot generalize to unseen sensors with different spectral dimensions. To address this limitation, **channel-invariant** approaches have been proposed. These include projection-weight interpolation (Sumbul et al., 2025; Varga et al., 2023), and channel-adaptive projection layers (Xiong et al., 2024; Wang et al., 2025; Li et al., 2025b). Several methods also exploit known wavelength information (for example via wavelength positional encodings) to establish cross-sensor channel relationships; we refer to such methods as **wavelength-aware** (Xiong et al., 2024; Wang et al., 2025; Li et al., 2025b; Sumbul et al., 2025; Varga et al., 2023). When a model is both channel-invariant and wavelength-aware, we call it **camera-agnostic**, since it can in principle generalize to any spectral sensor, whether seen during training or not.

Most camera-agnostic designs handle spectral inputs only at the projection stage and then perform purely spatial operations in feature space to learn spatial structure Xiong et al. (2024); Wang et al. (2025); Sumbul et al. (2025); Waldmann et al. (2025); Jakubik et al. (2025); Fuller et al. (2023); Tseng et al. (2025); critically, they do not learn spectral relationships within that feature space. We categorize them into **spatial encoding schemes** By contrast, **spatio-spectral encoding schemes** learn joint spatial *and* spectral relations in the feature space. This capability can be essential in spectrally heterogeneous settings where the model must align spectral signatures across different sensors. Examples include spatio-spectral patching with self-attention (Hong et al., 2024) or dedicated spectral encoders (Braham et al., 2024). However, existing spatio-spectral encoding schemes are not camera-agnostic.

To address this gap, we propose CARL—to the best of our knowledge, the first approach to unify explicit spatio-spectral feature encoding with a camera-agnostic design. CARL achieves this by introducing a channel-invariant, wavelength-aware spectral encoder that compresses variable-length channel inputs into fixed-length spectral representations, enabling the model to learn rich joint spatio-spectral features and to generalize strongly out-of-distribution without per-camera retraining.

**Self-supervised learning strategies for spectral imaging**   Self-supervised learning (SSL) is increasingly important as unlabeled data and compute scale up. Most current spectral-imaging pipelines inherit spatial encoding schemes from RGB models and therefore apply SSL that only learns spatial relations. We call that **spatial self-supervision**. These methods typically adapt RGB SSL recipes (e.g., MAE, DINOv2, iJEPA) to remote-sensing foundation models (He et al., 2022; Oquab et al.,

Table 9: **CARL learns robust representations during pre-training.** Linear-probing mIoU results on three evaluation datasets. Two of the datasets are acquired by sensors unseen during pre-training (Beijing, LoveDA). CAR shows competitive in-distribution performance and superior generalization to out-of-distribution sensors, achieving the best average rank across all 11 benchmark datasets.

| Sensor | SegMunich (mIoU) | Beijing (mIoU) | LoveDA Rural (mIoU) | Avg. Rank over all 11 datasets |
| --- | --- | --- | --- | --- |
| | Sentinel-2 (10 bands) | Gaofen-5 (116 bands) | RGB (3 bands) | |
| SpectralGPT | 27.9 | - | - | 5.5 |
| Galileo | 35.3 | - | - | 5.5 |
| Croma | - | - | - | 5.0 |
| DOFA | 38.2 | 17.4 | 14.9 | 3.2 |
| Copernicus-FM | 38.4 | 14.5 | 17.6 | 2.6 |
| SMARTIES | **39.1** | 17.1 | 17.3 | 2.6 |
| CARL | 38.9 | **19.1** | **29.8** | **1.6** |

2023; Assran et al., 2023; Fuller et al., 2023; Jakubik et al., 2025; Xiong et al., 2024; Waldmann et al., 2025; Tseng et al., 2025). In contrast, **spatio-spectral self-supervision** learns spatio-spectral relations during pretraining. For example, SpectralGPT performs reconstruction of 3D spatio-spectral patches (Hong et al., 2024). Scaling pretraining requires camera-agnostic pretraining, but this is not always feasible: some approaches require decoder heads tied to specific channel counts, which prevents straightforward cross-sensor pretraining (Sumbul et al., 2025). Moreover, SpectralGPT as backbone is not camera-agnostic. Additionally, pixel-reconstruction objectives are particularly sensitive in spectral imagery because pixel noise from atmospheric effects, illumination variation, and sensor distortions are stronger than in RGB data. For this reason, feature-based SSL methods (e.g., I-JEPA (Assran et al., 2023), DINOv2 (Caron et al., 2021)) — which learn robust latent representations rather than raw pixel reconstructions — may be better suited to spectral data

To the best of our knowledge, there is currently no SSL strategy that is both camera-agnostic *and* spatio-spectral, nor a strategy that is both feature-based *and* spatio-spectral. In this work we introduce CARL-SSL: a feature-based, camera-agnostic spatio-spectral self-supervision framework. CARL-SSL enables scalable pretraining across heterogeneous sensors and yields robust joint spatio-spectral representations.

## C  ADDITIONAL REMOTE SENSING EXPERIMENTS

In addition to Tabs. 4, 5, we provide further results on our linear probing evaluation in Tab. 9. In particular, we evaluated segmentation performance on the three land cover datasets. While SegMunich is acquired by the Sentinel-2 sensor, and is therefore in-distribution, Beijing and LoveDA were acquired by unseen sensors (hyperspectral and RGB), and therefore emphasizes on cross-generalizability. Further details to the datasets can be seen in Tab. 8. While every model except of Croma can be applied on the SegMunich images that have 10 channels, only three baseline methods can process hyperspectral images. Particularly in those scenarios, CARL's outperformance is pronounced. Furthermore, we report the average rank over all eleven datasets of all baseline methods. CARL ranks best with an average rank of 1.6.

As the SegMunich is sufficiently large for full fine-tuning, we performed supervised fine-tuning. Due to the required compute, we limit the baseline methods to DOFA, and SpectralGPT$^+$. As can be seen in Tab. 10, CARL keeps its outperformance and achieves a score of 50.9.

## D  COMPUTATIONAL COMPLEXITY

Encoding high-dimensional spatio–spectral data imposes computational and memory demands; therefore, controlling excessive floating-point operations (FLOPs) is essential. For the models considered in Sec. 4.3, we report parameter counts and FLOP estimates for the input sizes used in our remote-sensing evaluation. All experiments use the base parameter size of each model to provide a fair comparison across architectures. Two architectures, Galileo and Croma, were excluded from this complexity study because they were specifically designed for Sentinel-2 and cannot be reasonably adapted to other sensors without substantial redesign. By contrast, SpectralGPT depends on the

number of channels only through a single linear layer, making it straightforward to adapt to different sensor channel counts; accordingly, we include it in our study. The remaining models are sensor-agnostic and can be applied for the different input size out of the box. To make comparisons consistent, we fixed a patch grid of $16 \times 16$ and evaluated four channel configurations corresponding to RGB, Sentinel-2, OHS, and Gaofen-5 sensors (3, 12, 32, and 116 channels, respectively, see Tab. 8). We report FLOPs for multiple input sizes because the models scale differently with spectral dimensionality. Finally, we analyze how computational complexity relates to model design choices—such as encoding schemes—and to average model performance, to highlight trade-offs between efficiency and accuracy

In Tab. 11 we summarize the measured costs. Overall, purely spatial encoding schemes are computationally cheaper than spatio–spectral approaches; the differences in scaling behavior explain most of this gap. SpectralGPT uses 3D patching and full self-attention across both spatial and spectral dimensions for twelve blocks. Consequently, the complexity of each attention block grows as $\mathcal{O}((H \cdot W \cdot C)^2)$.

CARL takes a different design that yields substantially more favorable scaling by exploiting two principles: (1) disentangled spectral and spatial encoding, and (2) cross-attention with K learned spectral tokens. Concretely, CARL first extracts rich features along the channel axis, aggregates them, and then applies spatial-only transformer blocks. This produces spectral encoding stages whose attention cost scales as $\mathcal{O}(C^2)$ (four blocks), and spatial-only stages whose complexity scales as $\mathcal{O}((H \cdot W)^2)$ (eight blocks). Beyond lower FLOPs, disentanglement also simplifies spectral- and spatial-specific design choices (e.g., wavelength positional encoding) while preserving rich spatio-spectral representations. To further reduce costs, CARL interleaves a small number of full spectral self-attention operations with cheaper cross-attention blocks inside the spectral encoder. While spectral self-attention scales as $\mathcal{O}(C^2)$, a cross-attention over $K$ learned spectral representations scales as $\mathcal{O}(C \cdot K)$. Since $K$ is fixed and small in our experiments (we use $K = 8$), these cross-attention blocks are much less expensive than full spectral self-attention when $C$ is large (as in hyperspectral imagery). Although CARL remains more expensive than purely spatial encoders, the additional cost is offset by substantially richer features and markedly better out-of-distribution generalization. This trade-off is reflected in CARL's strong empirical performance (average rank 1.6 in our evaluation; see Tab. 11).

Because CARL is camera-agnostic, we can exploit this property to reduce training cost using a simple channel-subsampling strategy. For the hyperspectral organ dataset (originally 100 channels), we randomly sampled 16 channels at each training step and optimized the model on this reduced input. This approach reduced training FLOPs by roughly $\sim 75\%$ in our experiments while preserving validation performance: we obtained a mIoU of 68.8, compared to the original 69.1.

Two points explain why subsampling works well here. First, CARL's spectral encoder explicitly learns relationships across channels and compresses spectral information into a fixed set of representations. Therefore, the model can integrate information across different sampled subsets and still recover rich spectral features. Second, random channel sampling acts as a form of stochastic regularization: by seeing many different channel subsets during training, the model becomes more robust to missing or shifted spectral bands and generalizes better to out-of-distribution sensors. Practically, channel subsampling is an easy-to-implement, low-overhead augmentation.

# E  URBAN SCENE SEGMENTATION

The detailed class-wise IoU scores on the HSICity test set are depicted in Tab. 12. CARL-SSL exhibited superior performance benchmarked against camera-specific and channel-invariant spectral imaging models. As the HSICity training set does not contain any "pole" annotation, the camera-specific model exhibits a "pole" IoU score of 0. Notably, CARL and CARL-SSL achieved the best IoU scores for the "pole" class, indicating superior capability of translating RGB labels from Cityscapes to hyperspectral imagery.

Table 10: **Fully fine-tuned CARL yields superior downstream performance.** Class-wise IoU scores with 95 % confidence intervals on the SegMunich multispectral land cover dataset were benchmarked against strong SSL-pretrained models.

| Method | Arable land | Perm. Crops | Pastures | Forests | Water | Shrub | Open spaces | Wetlands | Mine. dump | Art. veg. | Urban fabric | Buildings | mIoU |
|---|---|---|---|---|---|---|---|---|---|---|---|---|---|
| SatMAE | 72.3 | 15.8 | 49.7 | 81.9 | 74.0 | 13.6 | 27.5 | 41.2 | 37.9 | 18.3 | 65.3 | 50.6 | 45.7 |
| SMARTIES | 71.6 | 19.6 | 49.4 | 85.2 | 72.9 | 17.3 | 44.8 | 39.3 | 37.3 | 22.2 | 65.6 | 53.2 | 48.20 [47.6; 48.7] |
| SpectralGPT$^+$ | 72.2 | 21.9 | 51.0 | 86.3 | **76.5** | 17.2 | 44.0 | 39.0 | 38.2 | 22.9 | 66.8 | 53.3 | 49.1 [48.6; 49.6] |
| DOFA | 72.0 | 21.5 | 50.7 | 86.1 | 75.3 | 18.2 | 45.4 | 40.1 | 39.1 | 23.1 | 67.3 | **54.9** | 49.5 [48.9; 49.9] |
| CARL | **72.9** | **24.9** | **51.9** | **86.6** | **76.5** | **21.0** | 43.9 | **42.0** | **42.0** | **25.6** | **68.6** | 54.8 | **50.9** [50.4; 51.4] |

Table 11: **Computational complexity of compared models.** For each sensor in our remote sensing evaluation, we report the number of parameters and estimated GFLOPs for a subset of models. "Avg. Rank" denotes the mean performance rank (lower is better). CARL offers a more favorable compute–performance balance than the spatio-spectral baseline SpectralGPT, while spatial-only encodings are cheaper but generalize poorly to unseen sensors.

| Model | Params (M) | GFLOPs (per sensor) | | | | Spatio-spectral | Avg. Rank |
|---|---|---|---|---|---|---|---|
| | | RGB | Sentinel-2 | OHS | Gaofen-5 | | |
| SpectralGPT | 85.4 | 23 | 107 | 339 | 2573 | ✓ | 5.5 |
| DOFA | 111.1 | 23 | 24 | 26 | 32 | ✗ | 3.2 |
| Copernicus-FM | 139.3 | 23 | 24 | 26 | 32 | ✗ | 2.6 |
| SMARTIES | 88.4 | 23 | 24 | 26 | 55 | ✗ | 2.6 |
| CARL | 71.5 | 34 | 53 | 96 | 286 | ✓ | **1.6** |

## F    SYNTHETIC MULTISPECTRAL DATA GENERATION

As outlined in Section 4.1 of the main manuscript, we synthesized multispectral images from given hyperspectral images to simulate spectral heterogeneity within the training set. This section provides a more in-depth explanation of the data generation procedure.

In spectral imaging, optical filters play a crucial role in isolating specific wavelength bands, allowing the system to capture reflectance information with spectral specificity, as discussed in (Garini et al., 2006). These filters are designed to selectively transmit light within defined wavelength ranges, determined by their material properties and design specifications. The transmission characteristics of an optical filter are typically described by its filter function, which quantifies the transmitted intensity as a function of wavelength. In practice, these filter functions often exhibit smooth, bell-shaped curves centered around a target wavelength (Niewiadomski, 2013).

To simulate this behavior in a proof-of-concept setting, we modeled the filter functions as normalized Gaussian distributions, where the mean corresponds to the center wavelength and the variance controls the bandwidth of the filter. This approach enables the generation of virtual multispectral cameras with tunable spectral profiles. Using these Gaussian-modeled filters, we synthesized multispectral images from hyperspectral data, while preserving the spatial context of the scene.

Specifically, we simulated a multispectral channel by first sampling the corresponding filter's center wavelength $\mu$ through farthest point sampling within [550 nm, 950 nm]. To define a realistic range for the filter bandwidth, we analyzed a real near-infrared multispectral camera (Ximea® MQ022HG-IM-SM5X5 NIR), which features 25 spectral channels. Particularly, we fitted Gaussian curves to the camera's filter functions to estimate plausible variance values. Based on this analysis, the variance of each simulated Gaussian filter was then uniformly sampled from the interval [5, 25]. As the given hyperspectral images exhibited wavelengths from 500 nm to 1,000 nm with 5 nm steps, we discretized the wavelength axis accordingly and set $\lambda = (500, 505, ..., 995)^T \in \mathbb{R}^{100}$. Then, we defined the

Table 12: **The proposed spectral encoder demonstrates superior performance as a camera-agnostic model.** The class-wise IoU scores with the 95 % confidence intervals of the mIoU scores on the HSICity test set. While the camera-specific model was pre-trained on Cityscapes and fine-tuned exclusively on HSICity, the other models are channel-invariant adaptations which were concurrently trained on both datasets. Notably, our spectral encoder performs best among the presented adaptation methods, and significantly benefits from self-supervised pre-training.

| | Camera-specific model | Spectral Adapter | HyperFree | Hyve | DOFA | CARL | CARL-SSL |
|---|---|---|---|---|---|---|---|
| Road | 93.4 | 93.6 | 93.7 | 94.0 | 94.1 | 94.7 | **95.0** |
| Sidewalk | 32.8 | 33.5 | 38.3 | 44.3 | 47.6 | 43.5 | **47.8** |
| Building | 69.8 | 54.9 | 65.0 | 69.4 | **71.9** | 71.1 | 71.1 |
| Wall | 55.1 | 43.9 | 54.7 | 54.4 | 52.4 | **55.4** | 55.2 |
| Fence | **14.1** | 5.3 | 11.6 | 11.5 | 10.4 | 11.3 | 13.9 |
| Pole | 0.0 | 29.6 | 15.1 | 20.8 | 30.9 | 31.0 | **31.8** |
| Traffic light | 51.0 | 50.4 | 47.8 | 53.0 | **58.8** | 55.5 | 57.2 |
| Traffic sign | 53.4 | 49.4 | 49.1 | 54.6 | 59.9 | 59.1 | **61.5** |
| Vegetation | 80.9 | 72.5 | 79.5 | 80.9 | **82.3** | 82.0 | 81.9 |
| Terrain | 3.2 | **9.0** | 5.7 | 3.6 | 3.9 | 6.6 | 5.1 |
| Sky | 85.8 | 79.9 | 83.7 | 87.2 | 88.5 | 88.4 | **88.7** |
| Person | 30.9 | 28.0 | 25.9 | **36.0** | 31.1 | 29.6 | 31.9 |
| Rider | 34.0 | 34.3 | 35.8 | 44.5 | 37.6 | 32.2 | 37.8 |
| Car | 86.0 | 88.1 | 86.2 | 88.0 | 89.5 | 89.3 | **90.3** |
| Truck | 53.7 | 50.8 | 57.0 | 60.6 | **73.6** | 57.5 | 63.0 |
| Bus | 67.6 | 80.0 | 79.3 | 80.3 | 83.0 | **87.6** | 87.4 |
| Train | - | - | - | - | - | - | - |
| Motorcycle | 0.0 | **0.4** | 0.0 | 0.0 | 0.1 | 0.0 | 0.0 |
| Bicycle | **36.0** | 11.3 | 19.2 | 29.1 | 26.5 | 29.1 | 33.1 |
| mIoU | 44.6 [40.9; 47.3] | 43.4 [41.0; 45.2] | 44.6 [42.2; 46.5] | 48.0 [45.4; 50.0] | 49.6 [46.8; 51.6] | 48.6 [45.6; 51.0] | **50.1** [47.2; 52.4] |

Table 13: **Semantic content drives CARL's feature variance.** Proportion of variance in the learned feature embeddings explained by semantic content (organ class) versus imaging sensor, as measured by mean $R^2$ across all embedding dimensions.

| Feature Variance Source | Explained Variance |
|---|---|
| Confounding Variable (Sensor) | 0.6 % |
| Semantic Content (Organ) | **61.6 %** |

L1-normalized filter function $\tilde{F}_{\mu,\sigma}$ as following:

$$F_{\mu,\sigma}(\lambda_i) = e^{-\frac{(\lambda_i - \mu)^2}{2\sigma^2}} \tag{9}$$

$$\tilde{F}_{\mu,\sigma}(\lambda_i) = \frac{F_{\mu,\sigma}(\lambda_i)}{\sum_{j=1}^{100} |F_{\mu,\sigma}(\lambda_j)|} \tag{10}$$

By uniformly sampling the number of channels, $C$, within $[10, 25]$, we obtained $C$ channel-specific filter functions, $(\tilde{F}_{i,\mu_i,\sigma_i})_{i \leq C}$, as described above. These functions can be collectively represented in matrix form as follows:

$$\tilde{F} = \begin{bmatrix} \tilde{F}_{1,\mu_1,\sigma_1}(\lambda)^T \\ \vdots \\ \tilde{F}_{C,\mu_C,\sigma_C}(\lambda)^T \end{bmatrix} \in \mathbb{R}^{C \times 100} \tag{11}$$

Finally, for a hyperspectral pixel $P_{HSI} \in \mathbb{R}^{100}$, we simulated the corresponding multispectral pixel by:

$$P_{MSI} = \tilde{F} \cdot P_{HSI} \in \mathbb{R}^C \tag{12}$$

This matrix-vector multiplication can be performed for each pixel, leading to an multispectral image with $C$ channels. Notably, we only altered spectral properties of the images, while preserving geometric information.

In this way, six camera configurations were simulated, resulting in six sets of synthetic multispectral images. To systematically introduce spectral heterogeneity into the original hyperspectral training data, a progressive substitution strategy with the synthetic multispectral images was employed. In each iteration, hyperspectral data from two additional porcine subjects was substituted with

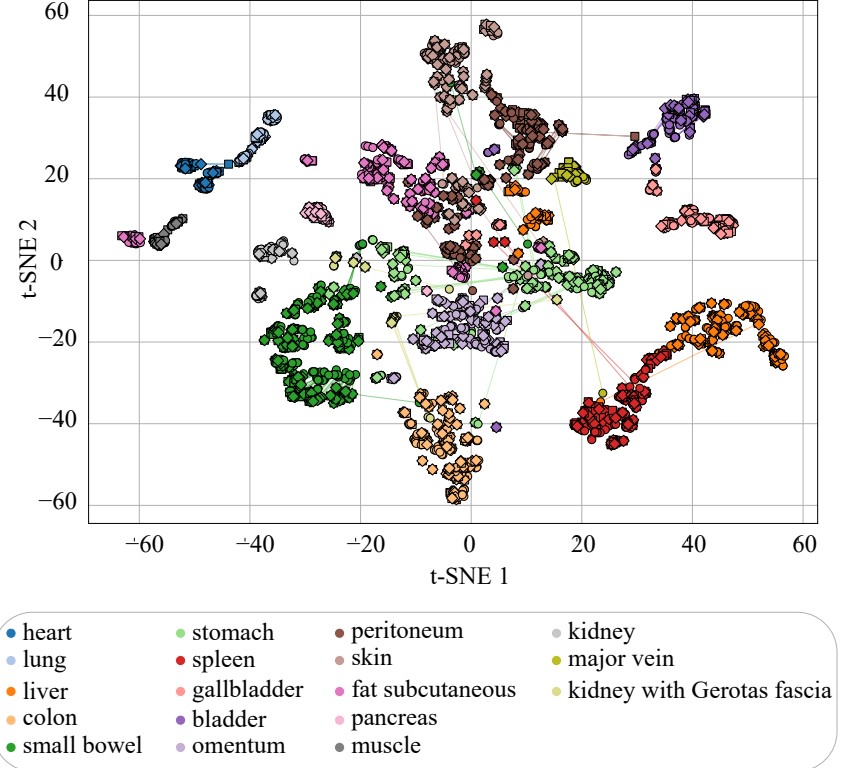

Figure 6: **CARL effectively disentangles organ semantics from camera variability.** t-SNE projection of mean feature embeddings for each organ region across the original hyperspectral image and two simulated multispectral variants. Points are color-coded by organ class and shaped by camera type (one hyperspectral, two unseen multispectral). Embeddings from the same region are connected into triangles. The dominant clustering by color and the scarcity of modality-connecting triangles demonstrate that the learned features are strongly organ-specific while remaining largely invariant to camera variations.

the corresponding synthetic multispectral images from a different simulated camera configuration. This process produced six augmented datasets that exhibit increasing spectral heterogeneity while preserving the surgical scene content. Model generalization to hyperspectral imagery was evaluated on the original hyperspectral test set, encompassing 166 images from five different porcine subjects.

## G ANALYSIS ON FEATURE REPRESENTATIONS

In Sec. 4.1, we generated spectrally heterogeneous training datasets while keeping the hyperspectral evaluation set unchanged to avoid any modifications during testing. We now extend the evaluation by adding two sets of synthetic multispectral images, generated via simulated filter responses as described in Sec. 4.1. For each hyperspectral sample in the evaluation set, we produced two corresponding multispectral versions. After training our model on a dataset variant from Sec. 4.1, we performed inference on this enlarged, spectrally diverse evaluation set. Importantly, the multispectral cameras from the evaluation set are unseen during training. For each labeled organ region, we computed the mean feature vector—obtained from the spatial encoder—over its ground-truth mask. This yielded three embeddings per region: one from the original hyperspectral image and two from the simulated multispectral variants. To visualize these embeddings, we applied t-SNE to project them into a two-dimensional space (Fig. 6). Embeddings corresponding to the same region but different modalities were connected to form a triangle. Marker shapes indicate camera type, while colors denote organ class. The strong color-based grouping, rather than marker-based grouping, demonstrates that our feature representations are organ-specific and largely camera-agnostic. This is further supported by the scarcity of visible triangles.

To quantify this observation, we performed a variance decomposition analysis using linear regression. Specifically, the feature embeddings were independently regressed against one-hot encoded predictors of either organ class or imaging sensor, and the coefficient of determination $R^2$ was computed for each regression. The average $R^2$ across all feature dimensions was then taken as the proportion of variance explained by the corresponding factor. As reported in Table 13, organ class explains 61.6 % of the feature variance, while the imaging sensor accounts for only 0.6 %. This indicates that the representations are strongly driven by semantic content while remaining largely invariant to the confounding sensor domain, suggesting robust disentanglement of task-relevant features from acquisition-specific artifacts.

## H  LLM USAGE STATEMENT

Large Language Models (LLMs) were used exclusively to assist with paper writing. Specifically, they were employed to correct grammar errors and enhance the clarity and style of existing text. Importantly, LLMs were not used to generate section drafts or even write entire sections. Their role was limited to refining and improving written material.

