# OpenReview forum: "CARL: Camera-Agnostic Representation Learning for Spectral Image Analysis"
_ICLR.cc/2026/Conference — ICLR 2026 Poster_

### Official Review · Reviewer_usvR · 2025-10-26

**Soundness:** 3
**Presentation:** 2
**Contribution:** 2
**Rating:** 4
**Confidence:** 4

**Summary:**

This paper introduces an architecture called CARL and a self-supervised learning (SSL) algorithm for it called CARL-SSL. CARL consists of a spectral encoder followed by a spatial encoder. The spectral encoder is essentially a transformer over the spectral dimension, i.e., a token is a spatial patch of a single spectral channel and attention occurs over tokens at the same spatial location (= inter-channel attention). The spectral encoder outputs K tokens per spatial location, which represent the spectral information (and some spatial information, since the patch size is not 1) at the location. The spatial encoder is a ViT.

CARL-SSL consists of two learning objectives. The first adapts VICReg to pre-train the spectral encoder and the second adapts I-JEPA to pre-train the spatial encoder.

The paper demonstrates CARL's superiority on many benchmarks and domains, such as remote sensing, medical imaging, and automotive data.

**Strengths:**

- The paper is dense with a lot of content, yet is well-written overall
- Both CARL and CARL-SSL are technically novel
- CARL is demonstrated across many domains
- The challenge that CARL aims to solve is important. Cameras with more colour channels contain more information that can be used to improve predictions, but the field has not settled on the optimal way to process many channels.

**Weaknesses:**

Major concern:
- Computational cost. It is not clear how much more or less expensive CARL is relative to baseline methods. I see a computational complexity section in the appendix, showing that for an input of 128x128x48, CARL is 15X the cost of DOFA. In my opinion, this makes the comparison unfair at this input size. The cost at other sizes is unclear. I believe that CARL is computationally expensive because it uses a separate spectral encoder and it may have more tokens, since it uses k=8 tokens per spatial location.

Moderate concerns:
- Limited novelty. The spectral encoder is a transformer over the spectral dimension, in a sense it reminds me of axial transformers/attention (which attend to one axis at a time). It does include learned query tokens, differentiating it from a purely axial framework, which means it always outputs K tokens per spatial location. Have the authors thought about using a spectral encoder with only self-attention, then pooling all tokens along the spectral axis? This is a simpler means of achieving K=1 tokens per spatial location.
- Limited ablations. The paper includes some ablations, e.g., Table 3c (a single result that removes the spectral SSL task) and Table 6 (spectral position encoding, aggregation, K, and the feature dimension). These are informative but I'd like some more fundamental ablations, e.g., removing the spectral encoder and simply tokenizing all spatio-spectral patches with a vanilla transformer.
- Benchmarks and Baselines. Although CARL is evaluated on many domains, within a domain the experiments are not rigorous enough, in my view. For example, in remote sensing (one of the primary domains of the paper), EuroSat and BigEarthNet are not used despite being the most popular benchmarks. Regarding baselines, the paper cites Galileo (SOTA RS foundation model) but does not compare to it.

**Questions:**

Please see weaknesses

---

> ### Author Response · Authors · 2025-11-24
> **Reviewer-Specific Response**
>
> We thank the reviewer for the comprehensive feedback and will integrate additional results and clarifications in the revised manuscript.
>
> >"Computational cost"
>
> We thank the reviewer for feedback. We dedicated the third section of the global rebuttal to this point. Additionally, we substantially improved the corresponding section in the revised manuscript by incorporating the suggestions.
>
> >"Limited novelty"
>
> We dedicated the first section of the global rebuttal to this. Moreover, we expanded the related work in our revised manuscript.
>
> >"Limited ablation"
>
> We thank the reviewer for the suggestions. We performed two targeted ablations to isolate the contributions of CARL’s spatio‑spectral design.
>
>
> #### **1. Spatio–spectral patch transformer**
>
> We implemented a vanilla transformer that takes spatio‑spectral patches. This variant resembles SpectralGPT but is adjusted to be channel-invariant. Conceptually, this design has three main disadvantages compared with CARL:
> - **Sensitivity to spectral resolution**: Patching along the spectral axis fixes a spectral patch size, so model behavior depends strongly on the input spectral resolution. This reduces flexibility when supporting RGB, multispectral, and hyperspectral inputs.
> - **Impaired wavelength encoding**: Merging multiple channels into a single spectral patch makes it difficult to apply precise wavelength positional encodings, particularly in scenarios of low spectral resolution.
> - **Less disentanglement of modalities**: CARL uses separate spectral and spatial encoders, which permits reuse of spatial SSL strategies and pretrained spatial checkpoints.
>
> **Quantitative results**:
> We duplicated our experiments on the medical dataset and measured mIoU for varying fractions of multispectral training data, while always validating on hyperspectral data.
>
> Fraction of multispectral data |1/2 |2/3 |1
> | -------| -------| -------| -------
> Spatio-spectral patch transformer (mIoU)| 25.2| 18.2| 4.4
> CARL (mIoU)| **64.5**| **64.2**| **66.4**
>
> #### **2. Axial‑attention variant**
>
> **We implemented an axial‑style variant that applies self‑attention only across channel tokens, followed by token pooling via summation**. We evaluated the adjusted model on spectrally heterogenous version of the medical data. The axial variant achieved 54.2 mIoU versus 66.4 mIoU for CARL, a 12.2‑point drop.
>
> **We attribute this gap to CARL's learned spectral pooling**. In supervised training the model implicitly acquire pooling behavior, but self‑supervised pretraining makes this capability explicit: CARL‑SSL is pretrained to predict masked channel tokens from abstract spectral representations, which forces the network to model inter‑channel spectral relations and to produce compact, informative spectral representations. As a result, CARL's learned pooling can selectively emphasize informative wavelength ranges rather than treating all channels equally, as sum‑pooling does.
>
> This design choice is consistent with prior analyses of hyperspectral data showing that the **intrinsic dimensionality (ID) of hyperspectral spectra is often much smaller than the number of measured bands**. For example, hyperspectral images with >100 bands commonly have an ID on the order of ~20 [1]. **Learning multiple spectral representations therefore helps the model focus on the true degrees of freedom in the spectral domain rather than on redundant measurements**.
>
>
> >"Benchmark & Baselines":
>
> We dedicated the second section of the global rebuttal to this raised concern. In particular, we added 7 datasets, including 4 unseen sensors, and 4 baseline methods.
>
>
> ---
>
> [1]: Robin, Amandine, et al. "Estimation of the intrinsic dimension of hyperspectral images: Comparison of current methods." IEEE Journal of Selected Topics in Applied Earth Observations and Remote Sensing 8.6 (2015): 2854-2861.

---

### Official Review · Reviewer_BT6d · 2025-10-28

**Soundness:** 3
**Presentation:** 3
**Contribution:** 3
**Rating:** 4
**Confidence:** 3

**Summary:**

The paper introduces a method for camera-agnostic spatio-spectral
representation learning across RGB, multispectral, and hyperspectral images. It
introduces a spectral encoder with self- and cross-attention to learn
camera-agnostic spectral representations, combined with I-JEPA spatial
pretraining for self-supervised learning. This approach is evaluated across medical
imaging, autonomous driving, and satellite imaging, and claims to outperform
both camera-specific and channel-invariant baselines in terms of generalization
to unseen sensors.

**Strengths:**

- The paper tackles a relevant problem: variability across spectral cameras that limits generalization.
 - The combination of spectral tokenization, spatio-spectral aggregation, and self-supervised pretraining is reasonable.
 - Demonstrates cross-domain evaluation, including medical, automotive, and remote sensing datasets.

**Weaknesses:**

- The methodological contribution is limited: the components (spectral tokenization, 2D token aggregation, spatial feature extraction, I-JEPA pretraining) are standard components. Presenting it as a new ``framework'' is a bit of an overstatement.
 - The claims ``first spatio-spectral camera-agnostic representation learning'' is overstated; I am not sure why a model such as DOFA does not fall into this category. The related work Panopticon [1] seems to provide a similar channel encoding scheme (any comment on this will be appreciated). A clear positioning with respect to other models such as Copernicus-FM [2], SMARTIES [3], and Fleximo [4] would also be useful.
 - The remote sensing evaluation is a bit weak. A comparison with Panopticon would be helpful. Regarding the claim ``our approach outperformed both camera-specific and channel-invariant baselines'', for an equivalent compute budget, it is not clear that the proposed model performs better than one trained for a single sensor or a fixed set of sensors. The experimental section does not demonstrate this. Moreover, pretraining is conducted on SpectralEarth and a Sentinel-2 dataset (only two sensors), and three out of the four datasets used for evaluation contain only Sentinel-2 data, which was already seen during pretraining.
 - The evaluation on GeoBench does not use all the available datasets. Moreover, only SpectralGPT, DOFA, and SatMAE are compared to on one of the datasets. To support the claim to outperform sensor-specific models, comparisons with Terramind [5], CROMA [6], and others would be useful. (It may be possible to reuse results reported from previous papers for some of the datasets).
 - Additionally, in Table 4, linear probing is used across three datasets. This is unusual for remote sensing segmentation tasks, where models typically use an UpperNet decoder or at least a few convolutions and upscaling layers. That could explain the relatively weak results.
 - Code is not provided at submission time.


[1] Waldmann, Leonard, et al. "Panopticon: Advancing any-sensor foundation models for earth observation." CVPR 2025.

[2] Wang, Yi, et al. "Towards a unified copernicus foundation model for earth vision." arXiv preprint arXiv:2503.11849 (2025).

[3] Sumbul, Gencer, et al. "SMARTIES: Spectrum-Aware Multi-Sensor Auto-Encoder for Remote Sensing Images." arXiv preprint arXiv:2506.19585 (2025).

[4] Li, Xuyang, et al. "Fleximo: A flexible remote sensing foundation model." arXiv preprint arXiv:2503.23844 (2025).

[5] Jakubik, Johannes, et al. "Terramind: Large-scale generative multimodality for earth observation." arXiv preprint arXiv:2504.11171 (2025).

[6] Fuller, Anthony, Koreen Millard, and James Green. "CROMA: Remote sensing representations with contrastive radar-optical masked autoencoders." NeurIPS 2023

**Questions:**

- How does CARL compare to Panopticon or Terramind on datasets not included in the pretraining corpus, specifically for true camera-agnostic generalization?
 - Why linear probing is used for segmentation tasks instead of standard decoders, and how this choice affects results?
 - How sensitive are results to pretraining dataset composition? For example, would adding more diverse sensors improve generalization?

---

> ### Author Response · Authors · 2025-11-24
> **Reviewer-Specific Response**
>
> Thank you — we appreciate the reviewer’s careful reading and constructive suggestions. Below we respond point-by-point and note how we addressed each issue in the revision.
> >"The methodological contribution is limited"
>
> We address novelty in detail in the first section of the global rebuttal.
>
> >"The claims first spatio-spectral camera-agnostic representation learning is overstated"
>
> We did not intend to overclaim. In the global rebuttal, we clarified the terminology, discussed related work, and highlighted our specific contributions. Additionally, the revised manuscript includes an expanded related work section.
>
> CARL consistently outperforms Panopticon across all levels of spectral variability. We attribute this improvement to the enhanced spatio-spectral encoding, as discussed in the first section of the global rebuttal.
> >"The remote sensing evaluation is a bit weak"
>
> To strengthen this, we added seven benchmark datasets (five from unseen sensors) and four additional competitive baselines, including two camera‑agnostic methods. Our remote‑sensing evaluation now covers 11 datasets and 6 baselines; data and results are summarized in the global rebuttal and integrated into the revised paper.
>
> >"The evaluation on GeoBench does not use all the available datasets"
>
> We incorporated the most relevant GeoBench datasets into the expanded evaluation, including m-bigearthnet and m-eurosat.
>
> >"Code is not provided at submission time."
>
> We apologize that we did not include the code in our submission. However, **we added pseudo-code of the forward passes of CARL and CARL-SSL to the appendix**. The python code will be published in the very near future.
> > "How does CARL compare to Panopticon or Terramind on datasets not included in the pretraining corpus, specifically for true camera-agnostic generalization?"
>
> We elaborated on both models in the first section of the global rebuttal.
>
> We would like to emphasize that Terramind is trained with fixed image projections for given sensors, and therefore does not generalize to unseen sensors.
>
> We tested Panopticon on the medical data with synthetic controlled spectral variability. This is achieved by replacing hyperspectral with synthetically generated multispectral data within the training set, while keep the hyperspectral validation set untouched. We obtain the following results:
> Fraction of multispectral data | 0 | 1/6 |1/3 |1/2 |2/3 |5/6 |1
> | -| -| -| -| -| -| -| -
> Panopticon| 60.0| 46.2| 57.5| 57.3| 59.8| 56.1 | 45.3
> CARL| **68.0**| **68.9**| **62.2**| **64.5**| **64.2**| **66.0**| **66.4**
>
> >"Why linear probing is used for segmentation tasks instead of standard decoders, and how this choice affects results?"
>
> We used linear probing on segmentation tasks to isolate pretrained feature quality. Replacing the linear probe with a UperNet head does not change the conclusions: CARL still outperforms alternatives. For example, on m‑SA‑crop‑type CARL improves from 26.5 mIoU (linear probe) to 30.8 mIoU with UperNet; DOFA improves from 21.7 to 27.3 — the relative advantage of CARL is preserved.

---

### Official Review · Reviewer_D1Nu · 2025-10-31

**Soundness:** 3
**Presentation:** 3
**Contribution:** 3
**Rating:** 8
**Confidence:** 3

**Summary:**

This paper introduces CARL, a novel framework for camera-agnostic representation learning across RGB, multispectral, and hyperspectral imaging modalities. The key innovation is a spectral encoder that uses wavelength positional encoding and learnable spectral representations with a self-attention-cross-attention mechanism to convert spectral images with arbitrary channel dimensionality into camera-agnostic representations. The authors also propose CARL-SSL, a self-supervised pre-training strategy combining spectral and spatial feature-based learning. Extensive experiments across medical imaging, autonomous driving, and satellite imaging demonstrate superior performance compared to camera-specific and channel-invariant baselines.

**Strengths:**

1. The paper addresses a critical bottleneck in spectral imaging—the inability of existing models to generalize across cameras with different spectral properties. This is particularly relevant for domains like medical imaging where sensor diversity is high.
2. CARL is the first approach to combine wavelength-awareness, channel-invariance, spatio-spectral encoding, and spatio-spectral SSL pre-training (Table 1). The wavelength positional encoding is an elegant solution for establishing cross-camera channel correspondences.
3. The evaluation spans three diverse application domains with both synthetic and real-world spectral variations. The progressive substitution experiments (Figure 5) provide compelling evidence of robustness to spectral heterogeneity.
4. CARL consistently outperforms strong baselines across all experiments. The cross-modality knowledge transfer demonstrated in Figure 4 (e.g., transferring "pole" labels from RGB Cityscapes to HSI) is particularly impressive.
5. The ablation studies (Table 6) validate key architectural decisions, and the variance decomposition analysis (Table 10) provides insight into what the model learns.

**Weaknesses:**

1.  While σ=3 and K=8 are validated through ablations, the paper doesn't discuss how sensitive these choices are across different domains or whether they need domain-specific tuning.
2. Scalability Limitations: The experiments use relatively modest model sizes (small and base versions). It's unclear whether the approach scales to larger foundation models or whether architectural modifications would be needed.
3.  While CARL-SSL shows improvements, the relative contribution of spectral vs. spatial self-supervision could be explored more thoroughly. The ablation in Table 3c is limited to a single dataset with reduced budget.

**Questions:**

1. The authors mention spatial resolution differences as a limitation. Could the wavelength positional encoding framework be extended to handle spatial resolution as metadata? Have you experimented with this?
2. How does CARL perform when test wavelengths fall outside the training distribution? For example, if trained on visible-NIR but tested on SWIR?
3. You mention sampling 32 channels during training when C>32. What is the impact of this sampling ratio? Could adaptive sampling based on wavelength diversity improve performance?

---

> ### Author Response · Authors · 2025-11-24
> **Reviewer-Specific Responses**
>
> Thank you — we appreciate the reviewer’s thoughtful feedback and careful questions.
>
> >"While σ=3 and K=8 are validated through ablations, the paper doesn't discuss how sensitive these choices are across different domains or whether they need domain-specific tuning."
>
> Our primary design goal was a spectral-imaging framework that works reliably out of the box across sensors and domains. To keep CARL broadly applicable, we avoided per-domain or per-sensor hyperparameter tuning and instead chose a compact set of defaults by running ablations on the medical dataset (which contains controlled spectral variations).
>
> To probe sensitivity, we ran a targeted ablation on SegMunich: training from scratch with K=8 versus K=16 spectral representations produced no meaningful difference in validation mIoU.
> This matches the saturation observed in the medical ablations in Table 6d: beyond a certain number of spectral representations, adding more yields no downstream benefit.
> This saturation is consistent with prior analyses showing that the intrinsic dimensionality of hyperspectral spectra is often much smaller than the number of measured bands (datasets with >100 bands commonly exhibit an ID on the order of ~20 [1]). In practice, a modest number of learned spectral representations is therefore sufficient to compress salient spectral information while keeping the method widely usable.
>
> >"The authors mention spatial resolution differences as a limitation". How to extend the framework with metadata?
>
> Satellite systems often provide metadata that supports cross-sensor generalization, including ground-sample distance (GSD) and per-channel spectral characteristics such as bandwidths. In practice, GSD can be incorporated as an image-level conditioning signal, while channel bandwidths can be naturally fused into the existing wavelength-encoding scheme. A practical limitation, however, is that detailed spectral metadata is often unavailable for many industrial spectral cameras, which hinders the development of metadata-aware models in certain application domains.
> >"How does CARL perform when test wavelengths fall outside the training distribution?"
>
> We probed this with an extreme cross‑wavelength experiment on the organ dataset. Training used only the first 50 of 100 channels (500–750 nm), while validation used the remaining 50 channels (750–1000 nm). Models trained from scratch exhibited a pronounced performance drop on the validation set.
>
> This shows that some overlap in wavelength coverage between training and evaluation is necessary for reliable cross‑sensor generalization. We did not attempt to determine the minimal required overlap in this study; doing so is a natural follow-up.
>
> These findings align with standard practice in spectral imaging: relevant wavelength ranges are typically chosen based on physical priors, and cameras are selected to cover those ranges. CARL is designed to mitigate gaps in channel count, bandwidth, and center wavelengths, but it cannot fully compensate when the test spectral content is fundamentally non‑overlapping with the training distribution.
>
> >"You mention sampling 32 channels during training when C>32. What is the impact of this sampling ratio? Could adaptive sampling based on wavelength diversity improve performance?"
>
> We elaborated on the channel subsampling in the global rebuttal in section on computional complexity.
>
> In addition, we also tested two sampling strategies: uniform sampling and farthest point sampling (FPS) based on the wavelengths. FPS selects a sparse set of channels that maximally covers the spectral range while retaining randomness. We ran the experiments on the hyperspectral organ data. As a results, both strategies reached comparable final accuracy, but FPS converged faster.
>
> ---
>
> [1]: Robin, Amandine, et al. "Estimation of the intrinsic dimension of hyperspectral images: Comparison of current methods." IEEE Journal of Selected Topics in Applied Earth Observations and Remote Sensing 8.6 (2015): 2854-2861.

---

### Official Review · Reviewer_YZcF · 2025-10-31

**Soundness:** 2
**Presentation:** 2
**Contribution:** 2
**Rating:** 4
**Confidence:** 5

**Summary:**

This paper proposes CARL (Camera-Agnostic Representation Learning), a framework designed to achieve cross-camera generalization across RGB, multispectral, and hyperspectral imaging systems. CARL introduces a spectral encoder with self- and cross-attention modules to map arbitrary channel-dimensional spectral images into a camera-agnostic latent space using wavelength positional encoding. The model further incorporates a spatio-spectral self-supervised learning (CARL-SSL) scheme combining spectral masking and spatial predictive learning (based on VICReg and I-JEPA). Experiments are presented across three domains, such as medical imaging, automotive vision, and satellite imaging, demonstrating improved robustness under heterogeneous spectral conditions and outperforming several baselines such as DOFA, Hyve, and SpectralGPT.

**Strengths:**

Problem relevance: The challenge of camera-specific spectral variability is real and significant in multisensor spectral imaging and remote sensing.

Comprehensive evaluation: The authors conduct cross-domain tests (medical, automotive, satellite), showing a broad view of potential applications.

Framework completeness: CARL integrates both wavelength-aware encoding and SSL, aligning with ongoing developments in spectral foundation models.

Potential impact: If scalable and reproducible, the framework could contribute to harmonized spectral representation learning across heterogeneous sensors.

**Weaknesses:**

Limited novelty / Incremental contribution.
Despite a well-written motivation, the technical innovation is incremental rather than groundbreaking. The proposed spectral encoder (self-attention + cross-attention) and wavelength positional encoding are straightforward extensions of existing works such as DOFA (Xiong et al., 2024), Hyve (Varga et al., 2023), and SpectralGPT (Hong et al., 2024). The SSL design (masked prediction + VICReg) follows existing feature-level pretraining frameworks (I-JEPA, DINOv2) without novel loss functions or theoretical insight. The “camera-agnostic” formulation is largely a rebranding of channel-invariant learning.

Lack of clear incremental improvement.
Reported gains (typically 1–2 mIoU or OA points) over existing baselines are marginal and fall within confidence intervals. The work does not convincingly demonstrate why the proposed cross-attention or wavelength encoding leads to generalizable representations beyond what existing channel-adaptive or wavelength-aware layers already achieve.

Insufficient discussion of existing multimodal or foundation models.
The manuscript lacks substantial comparison or discussion of recent multimodal foundation models (e.g., OmniSat, CROMA, Galileo, SeaMo, etc.) and fails to position CARL relative to these stronger baselines. This omission weakens its claim of being a “backbone for future spectral foundation models.”

Poor figure readability.
Figures (e.g., Fig. 1–5) are small, densely packed, and use fonts unreadable at standard scale. Important architectural elements and result visualizations are nearly illegible, making it difficult to evaluate the technical contributions.

Experimental design limitations.

The model is only tested on moderate-scale datasets (≤800 k samples) and not on true foundation-model-scale data.

Cross-domain transfer is limited to a few camera pairs (e.g., Cityscapes–HSICity); results on other heterogeneous sensors (e.g., PRISMA, ZY1-HSI) are missing.

Computational efficiency and scalability analyses are superficial, despite the model’s significant complexity.

Overstated claims.
The claim of being the “first camera-agnostic framework” is inaccurate, as previous models have already introduced similar wavelength- and channel-adaptive mechanisms. The novelty is therefore overstated.

**Questions:**

How does CARL differ quantitatively and conceptually from prior channel-invariant or wavelength-aware transformers?

What is the true scale of CARL in parameter count and computational cost compared to foundation models such as SpectralGPT or SatMAE?

Can the authors show clearer visualizations (e.g., spectral attention maps) to support claims of camera-agnostic learning?

How robust is CARL to non-spectral variations (spatial resolution, atmospheric noise, illumination)?

---

> ### Author Response · Authors · 2025-11-24
> **Reviewer-Specific Response**
>
> Thank you for the thoughtful feedback — we appreciate it and have incorporated it into the revised manuscript.
> > "Limited novelty / Incremental contribution."
>
> We addressed this in the first section of the global rebuttal and further clarified our contribution in a new expanded related-work section in the appendix.
>
> > "Lack of clear incremental improvement"
>
> The global response table highlights **substantial performance gains, especially on unseen sensors such as OHS, Gaofen-5, and LoveDA**. In the paper, Figure 5 shows a gain of more than 20 mIoU, clearly outperforming prior methods. Tables 3a, 3b, and 4 consistently report improvements greater than 5 mIoU. We also performed statistical tests to confirm significance. A recent study [1] surveyed median performance gains in medical image segmentation and, in that context, **the improvements we observe are on average 6.7 points higher**.
> > "Insufficient discussion of existing multimodal or foundation models"
>
> We expanded the discussion in the first section of the global rebuttal and added a dedicated section in the revised manuscript.
>
> >"Poor figure readability"
>
> Thank you for noting this. We have updated the figures in the revision to improve clarity and readability.
>
> >"The model is [...] not on true foundation-model-scale data."
>
> We elaborated on this in the global rebuttal (1/2). Scaling to foundation-model scale is a direction for future work.
> >"Cross-domain transfer is limited to a few camera pairs"
>
> **We strengthened the cross-sensor evaluation by adding five datasets acquired with unseen sensors**. The datasets and results are detailed in the global rebuttal and in the revised manuscript.
>
> >Computational efficiency analysis is superficial
>
> We added a complexity section in the global rebuttal and substantially revised the corresponding manuscript section to address this concern.
>
> >"Overstated claims"
>
> We did not intend to overstate our contributions. We have revised the claim language in the global rebuttal and in the new expanded related-work section to make our claims more precise and better supported.
>
> >"How does CARL differ quantitatively and conceptually from prior channel-invariant or wavelength-aware transformers?"
>
> We address this in the novelty section of the global rebuttal.
>
> >"What is the true scale of CARL in parameter count and computational cost"
>
> We incorporated this in the complexity discussion in the global rebuttal and clarified the model scale in the revised paper.
>
> >"Can the authors show clearer visualizations"
>
> Thank you for pointing this out.
> Using CARL's cross‑attention we extract attention maps over channel wavelengths, which reveal which spectral ranges each learned spectral representation attends to for a given task.
> This enhances the interpretability of CARL.
> We conducted this analysis on the organ dataset with controlled spectral variations.
> The attention maps showed that different spectral representations specialize on distinct wavelength ranges, and that the exact ranges depend on organ class.
> Crucially, **the focused wavelength patterns correlate with semantic differences between organ classes rather than with camera configuration, supporting the claim that CARL learns camera‑agnostic, class‑relevant features**.
> This finding is consistent with our feature analysis in Section G.
>
> >"How robust is CARL to non-spectral variations?"
>
> Thank you — this is an important point. Domain shifts in spectral imaging stem from heterogeneous spatial and spectral resolutions as well as environmental factors such as illumination and atmospheric effects. While some of these can be alleviated through augmentations, **spectral heterogeneity remains a key bottleneck due to inconsistent channel dimensionality**, which is the primary challenge our work addresses.
>
> **Our pretraining strategy prioritizes feature-level objectives that learn semantic spectral signatures rather than pixel-level appearance, improving robustness to environmental variation and sensor noise.** In contrast, pixel-wise reconstruction losses are more susceptible to such noise, often reducing downstream robustness.
>
> To further account for spatial-resolution differences, CARL can integrate a ground-sample-distance conditioning mechanism, enabling the model to remain aware of imaging scale.
>
> ---
>
> [1]: Christodoulou, Evangelia, et al. "Confidence intervals uncovered: Are we ready for real-world medical imaging AI?." International Conference on Medical Image Computing and Computer-Assisted Intervention. Cham: Springer Nature Switzerland, 2024.

---

### Author Response · Authors · 2025-12-02
**Global Rebuttal 2/2**

# Experiments
Although we performed a thorough evaluation across three different domains, which was intended to demonstrate the camera-agnostic nature of our approach, the reviewers requested additional evaluation, with a primary focus on remote sensing and earth observation data.

Accordingly, we strengthened our remote sensing linear-probing evaluation by adding the following datasets:
Dataset|Satellite|Channel Count|Wavelength Range (in nm)|Zero shot
-|-|:-:|:-:|:-:
m-bigearthnet (m-ben) [1]|Sentinel 2|12|442-2202|✗
m-eurosat [1]|Sentinel 2|13|442-2202|✗
Wuhan [2]|Gaofen-5|116|420-2400|✓
Beijing [2]|Gaofen-5|116|420-2400|✓
WHU-OHS [3]|Orbita Hyperspectral Satelite|32|466-940|✓
LoveDA [4]|Spaceborne|3|RGB|✓

Moreover, we also added several baselines methods (Croma, Galileo, Copernicus-FM, SMARTIES) in our evaluation and aggregated them into a single table, as shown below. For a more comprehensive comparison related to novelty, we grouped different methods according to their characteristics.
|Method|Spatio-spectral encoding|Camera-agnostic Encoding|Feature-based pretraining|m-ben|m-eurosat|m-forestnet|m-SA-crop-type|m-cashew|SegMunich|Wuhan|Beijing|LoveDA Rural|LoveDA Urban|Avg. Rank|
|-|:-:|:-:|:-:|-|-|-|-|-|-|-|-|-|-|-|
|SpectralGPT|✓|✗|✗|45.0|69.9|-|13.7|14.5|27.9|-|-|-|-|5.5|
|Galileo|✗|✗|✓|49.8|84.3|-|19.34|10.5|35.3|-|-|-|-|5.5|
|Croma|✗|✗|✓|59.5|86.6|-|25.15|12.1|-|-|-|-|-|5|
|DOFA|✗|✓|✗|61.0|89.9|43.8|21.7|18.2|38.2|20.3|17.4|14.9|12.6|3.2|
|Copernicu-FM|✗|✓|✗|62.1|87.2|44.8|26.5|14.5|38.4|18.1|14.5|17.6|15.4|2.6|
|SMARTIES|✗|✓|✗|62.0|92.6|49.8|24.3|12.7|39.1|18.8|17.1|17.3|13.5|2.6|
|CARL|✓|✓|✓|69.0|84.4|47.0|26.5|18.9|38.9|21.5|19.1|21.7|29.8|1.6|

CARL ranks first in a comprehensive evaluation across eleven datasets and five sensors—four of which were not seen during pretraining—and outperforms competitive remote sensing foundation models on eight datasets.
As shown in Tables 2 and 3 and Figure 5, CARL also delivers consistent improvements over camera-agnostic approaches such as DOFA across the other two application domains (medical and automotive).

We argue against the comment of reviewer YZcF that our improvements are marginal and just around 1-2 mIoU. The results in the table above show larger improvements and indicate the benefits of our methods on the remote sensing data, in addition to the other two application types presented in the paper.
# Computational Costs
We report GFLOPs for the input sizes used for the linear probing experiments:
|Model|Params (M)|GFLOPs (RGB)|GFLOPs (Sentinel-2)|GFLOPs (OHS)|GFLOPs (Gaofen-5)|Spatio-spectral encoding|Avg. Rank|
|-|-:|-:|-:|-:|-:|:-:|-:|
|SpectralGPT|85.4|23|107|339|2573|✓|5.5|
|DOFA|111.1|23|24|26|32|✗|3.2|
|Copernicus-FM|139.3|23|24|26|32|✗|2.6|
|SMARTIES|88.4|23|24|26|55|✗|2.6|
|CARL|71.5|34|53|96|286|✓|1.6|

SpectralGPT applies 3D patching and full self-attention over spatial and spectral dimensions for twelve blocks, giving each attention block a complexity of $\mathcal{O}((H\cdot W\cdot C)^2)$.

**CARL achieves more favorable scaling through two ideas: (1) disentangling spectral and spatial encoding, and (2) using cross-attention with K learned spectral tokens.**
It first extracts and aggregates features along the channel axis, then applies spatial-only transformer blocks whose cost is independent of $C$.
The spectral stages cost $\mathcal{O}(C^2)$ and spatial stages $\mathcal{O}((H\cdot W)^2)$.
To further reduce computation, CARL interleaves spectral self-attention with cheaper cross-attention over $K$ learned tokens, costing only $\mathcal{O}(C\cdot K)$; with $K=8$ this is much cheaper when C is large.
Although more expensive than purely spatial encoders such as DOFA, CARL produces richer representations and substantially stronger OOD generalization, reflected in its top empirical results (average rank 1.6).

To further reduce training costs, we leverage CARL’s channel invariance with channel subsampling during training. For the hyperspectral organ dataset (originally 100 channels), we randomly sampled 16 channels at each training step and optimized the model on this reduced input. **This approach reduced training FLOPs by roughly ~75% in our experiments while nearly preserving validation performance**: it achieved a mIoU of 68.8 compared to the original 69.1.

---

[1]: Lacoste, Alexandre, et al. "Geo-bench: Toward foundation models for earth monitoring."

[2]: Hong, Danfeng, et al. "Cross-city matters: A multimodal remote sensing benchmark dataset for cross-city semantic segmentation using high-resolution domain adaptation networks."

[3]: Li, Jiayi, et al. "WHU-OHS: A benchmark dataset for large-scale hersepctral image classification."

[4]: Wang, Junjue, et al. "LoveDA: A remote sensing land-cover dataset for domain adaptive semantic segmentation."

---

### Author Response · Authors · 2025-12-02
**Global Rebuttal 1/2**

We thank the reviewers for their constructive feedback, which has helped us improve the manuscript. We respectfully ask the area chair to read this global rebuttal, which addresses the reviewers' primary concerns; reviewer-specific points are discussed in the individual threads.
# Novelty
We were asked by several reviewers to provide a clearer explanation of our contribution and position our approach in relation to existing approaches. We focus on spectral camera images where the number of camera channels varies from sensor to sensor. The central challenge is twofold: **(1) make representations channel‑invariant** so a single model can be applied to many sensors, and **(2) make representations spectrally and spatially aware** so they capture the rich spatio‑spectral structure in the data. Existing approaches address only one of the issues, as outlined below.

Dealing with spectral information:
- **Channel-dependent projections**: Cannot accept arbitrary channel count due to fixed projections (Terramind, Croma, Galileo, OmniSat, SeaMo).
- **Channel-invariant projections**: Support arbitrary channel count (DOFA, Hyve, Copernicus-FM, SMARTIES, FlexiMo, Panopticon). For example, DOFA and FlexiMo leverage hypernetworks to generate projection weights; SMARTIES and Hyve interpolate projection weights.
- **Wavelength-aware projections**: Incorporate wavelengths into the projection (DOFA, Hyve, Copernicus-FM, SMARTIES, FlexiMo, Panopticon) for instance by conditioning a hypernetwork.
- **Camera-agnostic projections**: Channel-invariant + wavelength-aware (CARL, DOFA, Copernicus-FM, SMARTIES, Panopticon).

Dealing with spatial information:

After handling the spectral information through these projections, models exclusively perform spatial operations (e.g., ViT blocks) to learn spatial structure (e.g., DOFA, Hyve, Copernicus-FM, SMARTIES, Croma, Galileo, FlexiMo, Terramind, Panopticon). Importantly, they do not learn spectral relations within the feature space.
- **Spatio-spectral encoding**: Learn spatial and spectral relations in feature space via 3D patching (SpectralGPT) or dedicated spectral encoders (SpectralEarth). But existing spatio-spectral approaches are not camera-agnostic and require camera-specific retraining.
- **Spatio-spectral camera agnostic encoding**: To the best of our knowledge, **CARL is the first approach that unifies explicit spatio‑spectral feature encoding with camera‑agnostic design**. This combination lets CARL learn rich image representations, yielding strong generalization without per‑camera retraining.

To this end, CARL integrates a novel camera-agnostic spectral encoder with a spatial encoder. The spectral encoder features learnable spectral representations that learn salient spectral information through a self-attention-cross-attention mechanism. Channels of distinct cameras are linked with wavelength positional encoding. Finally, the spectral representations are forwarded to a ViT-like spatial encoder.
Through the fixed length spectral representations, the model is more robust against flexible channel counts.

**Our second contribution is SSL pretraining.**

Effective spatio‑spectral encoding at scale requires camera‑agnostic pretraining so we can leverage images from many sensors, and feature‑based SSL, offering higher robustness against sensor and atmospheric noise. CARL‑SSL addresses both needs by (1) proposing a **novel camera-agnostic feature-based spectral pretraining strategy** and (2) making it compatible with the spatial I-JEPA pretraining.

We detail related approaches below according to the SSL schemes they employ.
- **Camera-agnostic spatial SSL**: Only learn spatial relations in pretraining due to a spatial encoding scheme (Copernicus-FM, DOFA, SMARTIES, Panopticon)
- **Spatio-spectral SSL**: Learn spectral and spatial relations due to a spatio-spectral encoding scheme (SpectralGPT). SpectralGPT’s pretraining is neither channel invariant nor wavelength aware.
- **Camera-agnostic spatio-spectral SSL: CARL-SSL is the first method for spatio-spectral pretraining in a camera-agnostic design**. Moreover, CARL-SSL learns feature maps exclusively within the feature space.

Concretely, CARL-SSL learns compact spectral representations of a subset of unmasked channel tokens. A predictor then reconstructs the masked channel tokens from those abstract representations. To make the model camera‑agnostic and robust to different sensors, both the encoder and the predictor incorporate a wavelength positional encoding.

We would like to clarify one aspect: Although our model is well-suited as a backbone for foundation models—capable of generalizing across varied camera sensors—**we do not present a trained foundation model here**. The goal is to address data silos, enabling the reuse of annotated spectral data across evolving camera designs. Scaling to a foundation model will require data and compute, which we identify as important future work.

---

### Author Response · Authors · 2025-12-03
**Revised Manuscript**

Thank you — we incorporated the reviewers’ constructive feedback into a revised manuscript (changes are marked in blue). We again thank the reviewers for their time and helpful suggestions.

Below are the major and minor changes in this revision.

### **Major Changes**

- **Remote sensing experiments:**
Extended linear probing remote sensing results are reported in Tables 4, 5 and in Table 9 (appendix).
The previous Table 5 (full finetuning on SegMunich) is now in the appendix as Table 10 and has been extended with the SMARTIES baseline.
The benchmark datasets used for linear probing are listed in the new Table 8.

- **Computational complexity:**
Section D was substantially revised following reviewer feedback.
We now report parameter counts and FLOPs for the channel configurations used in the linear‑probing experiments (Table 11), add theoretical discussion to explain the models’ FLOP scaling behavior, and elaborate the channel‑subsampling strategy used to reduce compute.

- **Related work and foundation models:**
Following the reviewers’ suggestions, we created a new section on the expanded related work (Section B) to better position our approach relative to existing remote‑sensing foundation models, and clarified our specific contribution to avoid overstated claims.
We also performed little changes to the related work section of the main manuscript.

### **Minor Changes**
- Improved Figure 3 for better readability.
- Updated Table 1 to include the additional baselines used in the linear‑probing evaluation.
- Small text edits throughout to improve clarity while preserving the page limit.

---

### Meta-Review · Area_Chair_1TMk · 2026-01-11

**Summary:**

This paper proposes Camera-Agnostic Representation Learning (CARL), a framework designed to achieve cross-camera generalization across RGB, multispectral, and hyperspectral imaging systems. CARL introduces a spectral encoder with self- and cross-attention modules to map spectral images with arbitrary channel dimensionality into a camera-agnostic latent space via wavelength positional encoding. All reviewers raised valid concerns and constructive suggestions, and the authors provided detailed and persuasive responses. As a result, the main technical concerns, particularly those regarding computational complexity, comparison fairness, and experimental scope, have been sufficiently addressed in the rebuttal and revision. Some reviewers express mild reservations about the level of novelty, which cannot be fully resolved by additional experiments and clarifications alone. Overall, considering the reviews and the author response, AC believes the paper will provide valuable insights and a useful step toward robust multi-sensor spectral representation learning for the ICLR community, and therefore recommends Weak Accept (poster).

**Reviewer Concerns:**

Reviewer YZcF (Marginally below acceptance) raised concerns about limited novelty and potentially overstated claims, marginal gains, insufficient positioning vs. recent remote-sensing/foundation models (e.g., OmniSat/CROMA/Galileo/SeaMo, etc.), poor figure readability, and weak/scaled experimental evidence (limited sensors, not foundation-scale, superficial compute analysis, limited cross-sensor transfer). In response, the authors repositioned the novelty more carefully via a global rebuttal plus a new expanded related-work section, explicitly clarifying what is new (camera-agnostic and spatio-spectral feature encoding + camera-agnostic SSL) and toning down overclaims. They also substantially strengthened remote-sensing evaluation by adding multiple datasets (including unseen sensors) and additional baselines, aggregating results into a unified table and emphasizing larger gains on several datasets. This rebuttal should be able to resolve most concerns on the technical and experimental part.

Reviewer D1Nu (Accept) was strongly positive, viewing the work as addressing an important bottleneck with an effective design, and asked mainly about sensitivity/robustness (hyperparameters like σ and K across domains, spatial-resolution metadata, generalization to out-of-distribution wavelengths, and channel-subsampling strategy). The authors responded with targeted evidence and clarifications. They motivated the default hyperparameters as out-of-the-box choices and added a SegMunich sensitivity check, discussed metadata conditioning, provided an extreme cross-wavelength split experiment showing performance drops when train/test wavelength ranges do not overlap (thus clarifying limits), and elaborated on channel subsampling. Overall, the reviewer’s concerns were largely resolved with additional experiments and clearer framing of limitations.

Reviewer BT6d (Marginally below acceptance) centered on limited methodological novelty, overstated first-claim, and weak remote-sensing evaluation and baseline coverage (requested Panopticon/Terramind/CROMA, more GeoBench datasets, and clearer justification of linear probing for segmentation). The authors addressed these by (1) tightening novelty/positioning language and expanding related work, (2) broadening remote-sensing experiments substantially, (3) giving a direct Panopticon comparison under controlled spectral-variability settings, (4) clarifying that Terramind uses fixed sensor-specific projections and therefore cannot generalize to unseen sensors in their framing, (5) justifying linear probing as isolating feature quality and adding a UperNet-head example showing conclusions unchanged. These changes substantially mitigate the experimental/baseline and overclaim concerns.

Reviewer usvR (Marginally below acceptance) criticized computational cost fairness (CARL appeared far more expensive than DOFA at certain settings and that cost scaling across input/channel regimes was unclear), plus limited novelty (spectral transformer reminiscent of axial attention), insufficient ablations (e.g., remove spectral encoder; compare to simpler pooling-only designs; compare to spatio-spectral patch transformer), and remote-sensing benchmark/baseline gaps (EuroSat/BigEarthNet, lack of Galileo comparison). The authors responded by substantially revising the complexity analysis, and by running two key architectural ablations regarding channel-invariant spatio-spectral patch transformer variant and an axial self-attention + sum-pooling variant, arguing CARL’s learned spectral pooling + SSL objective is critical. They also expanded remote-sensing datasets and baselines, addressing most benchmark/baseline concerns.

**Reviewer Scores:**

Given the solid response and detailed evidences in rebuttal, the three reviewers YZcF, BT6d,  usvR are likely to raise the score to some extent. AC can not imagine a significant rise, since the concerns regarding technical novelty could not be completely resolved by adding more experiments. Reviewer D1Nu is quite positive. Given the concerns have been further addressed by the rebuttal, this reviewer would probably keep the score of accept unchanged.

---

### Decision · Program_Chairs · 2026-01-26

Accept (Poster)